# Neuroprotective role of Hippo signaling by microtubule stability control in *Caenorhabditis elegans*

**Hanee Lee, Junsu Kang[†], Sang-Hee Lee[‡], Dowoon Lee, Christine H Chung, Junho Lee***

Department of Biological Sciences, Institute of Molecular Biology and Genetics, Seoul National University, Seoul, Republic of Korea

*For correspondence: elegans@snu.ac.kr

Present address: [†]Department of Cell and Regenerative Biology, University of Wisconsin School of Medicine and Public Health, University of Wisconsin–Madison, Madison, United States; [‡]Korea Basic Science Institute, Ochang, Cheongju, South Korea

Competing interest: The authors declare that no competing interests exist.

## eLife Assessment

In their **valuable** study, Lee et al. explore a role for the Hippo signaling pathway, specifically wts-1/LATS and the downstream regulator yap, in age-dependent neurodegeneration and microtubule dynamics using *C. elegans* mechanosensory neurons as a model. The authors demonstrate that disruption of wts-1/LATS leads to age-associated morphological and functional neuronal abnormalities, linked to enhanced microtubule stabilization, and show a genetic connection between yap and microtubule stability. Overall, the study employs robust genetic and molecular approaches to reveal a **convincing** link between the Hippo pathway, microtubule dynamics, and neurodegeneration.

## Abstract

The evolutionarily conserved Hippo (Hpo) pathway has been shown to impact early development and tumorigenesis by governing cell proliferation and apoptosis. However, its post-developmental roles are relatively unexplored. Here, we demonstrate its roles in post-mitotic cells by showing that defective Hpo signaling accelerates age-associated structural and functional decline of neurons in *Caenorhabditis elegans*. Loss of *wts-1*/LATS, the core kinase of the Hpo pathway, resulted in premature deformation of touch neurons and impaired touch responses in a *yap-1*/YAP-dependent manner, the downstream transcriptional co-activator of LATS. Decreased movement as well as microtubule destabilization by treatment with colchicine or disruption of microtubule-stabilizing genes alleviated the neuronal deformation of *wts-1* mutants. Colchicine exerted neuroprotective effects even during normal aging. In addition, the deficiency of a microtubule-severing enzyme *spas-1* also led to precocious structural deformation. These results consistently suggest that hyper-stabilized microtubules in both *wts-1*-deficient neurons and normally aged neurons are detrimental to the maintenance of neuronal structural integrity. In summary, Hpo pathway governs the structural and functional maintenance of differentiated neurons by modulating microtubule stability, raising the possibility that the microtubule stability of fully developed neurons could be a promising target to delay neuronal aging. Our study provides potential therapeutic approaches to combat age- or disease-related neurodegeneration.

## Introduction

Neurons perceive extracellular signals and transmit the information to other cells, often over long distances. For this to happen, neuronal cells extend their axons and dendrites from 1 μm to more than 1 m and must maintain an intact morphology throughout their lifetime. The long, elaborate neuronal structures are vulnerable to degeneration associated with organismal aging or pathological conditions. Structural abnormalities and the consequent functional decline are hallmarks of aged neurons

or pathological neurons affected by neurodegenerative diseases (*Yankner et al., 2008*; *Bishop et al., 2010*; *Fjell and Walhovd, 2010*). Several studies have shown that the aged brain exhibits structural abnormalities such as synaptic loss, neuronal sprouting and restructuring, rather than neuronal cell death (reviewed in *Yankner et al., 2008*).

Microtubules form essential cytoskeletal structures for most eukaryotic cells. In neurons, they serve as structural struts that allow the neuron to develop and maintain a specific shape (*Barnes and Polleux, 2009*; *Marin et al., 2010*). They also act as railroads for motor proteins that deliver cellular cargo from cell bodies to distal axons (*Hirokawa et al., 2010*; *Maday et al., 2014*), as well as seeds for microtubule polymerization (*Job et al., 2003*). Microtubules of varying lengths are observed in neurons, and they can be both stable and dynamic. Numerous studies have suggested that both hyper- or unstable microtubules are harmful for maintaining neuronal morphology as the imbalance in microtubule dynamics contributes to neurological diseases (reviewed in *Dubey et al., 2015*). Thus, microtubule regulators have been implicated in a number of human diseases (*Dubey et al., 2015*; *Matamoros and Baas, 2016*) and signaling pathways including Notch pathway have been proposed as pharmaceutical targets for cytoskeletal protection of postnatal neurons (*Bonini et al., 2013*). Microtubule dynamics in normal neuronal aging, on the other hand, are poorly understood at the organismal level.

The neurons of *Caenorhabditis elegans* provide a valuable model system because they share fundamental characteristics with mammalian neurons in terms of their molecular basis and functions. Major genetic pathways governing early neuronal development such as neuronal cell migration or polarization were initially identified from numerous studies on *C. elegans* (*Hedgecock et al., 1990*; *Wadsworth et al., 1996*; *Culotti and Merz, 1998*; *Ikegami et al., 2004*). Additionally, *C. elegans* neurons could be utilized to understand the molecular mechanism of neuronal aging as they demonstrate age-associated structural and functional deterioration. In particular, touch receptor neurons (TRNs), comprising two ALM, AVM, two PLM, and PVM, show the most prominent decline in structure and function with aging (*Pan et al., 2011*; *Tank et al., 2011*; *Toth et al., 2012*). In aged animals, TRNs have highly wavy or ectopically branched processes, misshaped cell bodies with abnormal morphology or outgrown processes. Moreover, *C. elegans* exhibits gradual decreases in touch responses as they age (*Tank et al., 2011*). DAF-2/IGF pathway, JNK/MAPK pathway, and membrane activity of TRNs have been identified to influence the speed of touch neuronal aging by modulating whole organismal senescence or cell-autonomous functions (*Pan et al., 2011*; *Tank et al., 2011*; *Toth et al., 2012*).

The Hippo (Hpo) kinase signaling pathway, which is evolutionarily conserved across animal species, plays a pivotal role in the regulation of tissue size homeostasis (*Harvey et al., 2013*). It regulates cell proliferation and apoptosis during early development and tumorigenesis. When the Hpo pathway is active, LATS, the core kinase of the pathway, phosphorylates and inhibits the nuclear localization of YAP, a transcriptional co-activator. In the nucleus, YAP regulates target gene expression through interaction with the TEAD transcription factor (*Zhao et al., 2007*; *Zhao et al., 2008*). The dysregulation of this pathway promotes uncontrolled cell proliferation, leading to tumorigenesis and developmental problems (*Huang et al., 2005*; *Zhao et al., 2007*; *Harvey et al., 2013*; *Zheng and Pan, 2019*). In *C. elegans*, core components and their genetic interaction of the Hpo pathway are conserved. Inhibition of YAP-1/YAP by upstream WTS-1/LATS is needed for the maintenance of apicobasal membrane polarity of the developing intestine (*Kang et al., 2009*; *Lee et al., 2019*). The functions of the Hpo pathway in growth control are extensively established; however, its post-developmental functions are relatively unknown.

Here, we establish a novel genetic animal model that reveals the postnatal roles of the Hpo pathway in the differentiated neurons. Loss of *wts-1* leads to premature decline of TRNs both in structure and function, and downstream effectors, *yap-1* and *egl-44*, are required in this phenomenon. We employ genetic and chemical approaches to elucidate the cellular and molecular mechanisms by which *wts-1*-deficient TRNs lead to a premature deformation. Our results demonstrate that hyper-stabilized microtubules of the *wts-1* mutant or aged organism are responsible for neuronal deformation. The fact that loss of *spas-1*, a putative microtubule-severing enzyme, accelerates the onset of age-associated neuronal deformation also supports the contribution of hyper-stabilized microtubules in neuronal failures. Our study unveils the post-developmental functions of the Hpo pathway in the maintenance of neuronal integrity and highlights the importance of neuronal microtubule status in cellular aging and clinical approaches.

## Results

### *wts-1* mutant showed impaired structure of TRNs

A previous study demonstrated that the intestinal Hpo pathway is essential for the proper development of the intestinal lumen and loss of *wts-1*, the core kinase of the pathway, leads to early larval death (*Lee et al., 2019*). To define the non-intestinal roles of the Hpo pathway in *C. elegans*, we used the mosaic *wts-1* mutant system of which larval lethality was overcome with the intestine-specific expression of WTS-1. Since *wts-1* activity in all tissues except the intestine is absent in this system, hereafter we will refer to it as the *wts-1* mutant shortly.

We observed that the *wts-1* mutants showed extremely distinctive abnormalities in the morphology of TRNs. In the wild-type, ALM and its posterior homolog PLM extended their neuronal processes along the anterior–posterior axis and maintained structural integrity during adolescence (*Figure 1A and B*). However, the *wts-1* mutant- ALM and PLM exhibited ectopic swelling/waviness and branching (*Figure 1A and B*), and other various deformations, including extended or shortened distal processes or outgrowth of processes from the cell body (somatic outgrowth) (*Figure 1—figure supplement 1A and B*). Atypical structures of other TRNs, AVM and PVM, were also found in the mutants (*Figure 1— figure supplement 1C and D*), indicating that *wts-1* is essential in mechanosensory neurons. As ALM and PLM defects were much more severe, frequent, and easier to observe, we focused on these neurons for further study.

TRNs of *C. elegans* develop rapidly in their early life and govern escape behaviors in response to aversive stimuli (*Chalfie et al., 1985*). ALM and PLM complete their development during embryogenesis, whereas AVM and PVM develop until the first larval stage (L1) (*Chalfie et al., 1985*). To determine whether *wts-1* is required for the development or maintenance of TRNs, neuronal morphology of early larva was analyzed. The *wts-1* mutant had intact, unaltered neuronal processes of ALM and PLM at the very early L1 stage and the frequency of abnormal neurons gradually increased as the worms grew (*Figure 1C and D*). At the fourth larval (L4) stage, 98.89% of ALM and 84.44% of PLM, respectively, showed at least one morphological abnormality in the mutant, whereas they maintain intact structures in the wild-type (*Figure 1D*). Among the abnormalities, ectopic swelling and branching on the processes were the most common (*Figure 1—figure supplement 1E and F*), followed by extension of the distal process and somatic outgrowth of the cell body (*Figure 1—figure supplement 1G and H*). All these structural deformations were gradually increased. Ectopic swelling or branching occurred in multiple places in a single process and the number of swelling/branching also increased as the worms grew (*Figure 1—figure supplement 1I*).

To determine whether *wts-1* acts in the structural maintenance of neurons generally or TRNs specifically, we analyzed the morphologies of other neurons in the mutants. Dopaminergic, GABAergic, and cholinergic neurons were found to maintain structures comparable with those of the wild-type at the same age (*Figure 1—figure supplement 1J and K*). It is noteworthy that ALN, a cholinergic neuron that extends its process in association with ALM (*White et al., 1986*), preserved their structural integrity in the mutants (*Figure 1—figure supplement 1J*). Thus, we concluded that the loss of *wts-1* leads to a wide range of structural deformities specifically in TRNs and that this deformity is a matter of maintenance, rather than development of the structures.

### *wts-1* deficiency causes TRNs to deteriorate its structure and function

TRNs govern the aversive movements of worms in response to gentle body touches (*Chalfie et al., 1985*). To determine whether structurally deformed TRNs of *wts-1* affect neuronal functions, we measured the gentle touch responses of the mutant. At the L4 stage, control worms responded to almost every gentle touch (9.40 responses/10 touches), whereas *wts-1* mutants responded to 60% of stimuli (6.31/10) (*Figure 1E*). On the first day of adulthood (1DA), control worms showed slightly decreased touch responses compared with the L4 stage (8.25/10), whereas *wts-1* mutants displayed more impaired responses (5.57/10) (*Figure 1E*).

The structural and functional decline in the *wts-1* mutants is very similar to the phenotypes of normally aged animals. Morphological alteration and consequent functional decline are typically observed in the aged nervous system across the animal kingdom (*Yankner et al., 2008*; *Bishop et al., 2010*; *Fjell and Walhovd, 2010*). Aging of human brain in the absence of disease is often accompanied by structural deformation, such as dendritic restructuring, neuronal sprouting, and synaptic losses, rather than severe neuronal degeneration or cell death (*Yankner et al., 2008*). Consistently, the

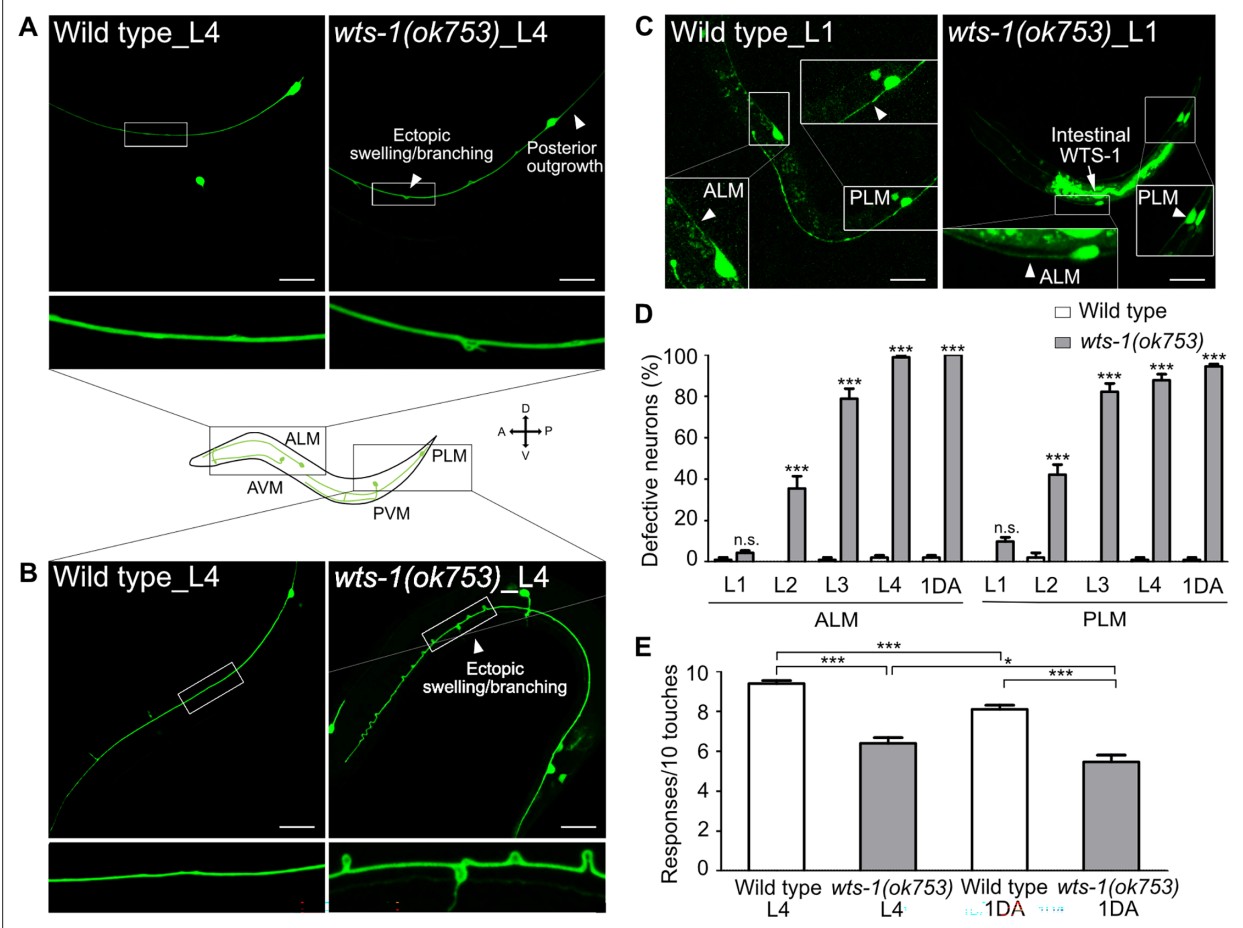

**Figure 1.** *wts-1* mutant shows the premature structural and functional decline of touch receptor neurons (TRNs). (**A, B**) Representative images of TRNs (**A**: ALM; **B**: PLM) in the wild-type and *wts-1(ok753)* mutant at the L4 stage. In the mutant, morphologically disrupted ALM and PLM were observed to exhibit abnormalities including ectopic swelling/branching on the processes and posterior outgrowth (arrowheads). (**C**) Newly developed ALM and PLM in the wild-type and the *wts-1(ok753)* mutant at the early L1 stage. The intact neuronal process is indicated using arrowheads. Intestinal expression of the WTS-1 rescue construct (P*opt-2*::WTS-1::GFP) is indicated using an arrow. (**A–C**) TRNs were visualized by expressing GFP under the control of the *mec-7* promoter (*muIs35*[P*mec-7*::GFP]) and anterior is to the left unless otherwise noted. Scale bar = 20 µm. (**D**) Penetration of defective TRNs of the wild-type and the *wts-1(ok753)* at different developmental stages. Neurons displaying any morphological abnormalities such as swelling, branching, somatic outgrowth, and extended distal process were scored as defective neurons. In one experiment, 30 cells were observed for each strain and each stage and the experiments were repeated three times. Statistical significance was determined by a two-way ANOVA followed by Bonferroni's post-test. (**E**) Quantified touch responses of the wild-type and the *wts-1(ok753)* mutant at L4 and first day of adulthood (1DA). N = 30. Statistical significance was determined by an unpaired *t*-test. ***p<0.001, **p<0.01, *p<0.05, n.s., not significant, compared to WT or control, unless otherwise marked on the graph. All data are presented as means ± SEM, unless otherwise noted.

The online version of this article includes the following source data and figure supplement(s) for figure 1:

**Source data 1.** Raw data for panels D and E.

**Figure supplement 1.** Loss of *wts-1* leads to touch neuronal-specific structural decline.

**Figure supplement 1—source data 1.** Raw data for panels D–I and K.

*C. elegans* nervous system exhibits age-associated structural deformation. Among several neurons, TRNs exhibited the most robust decline in the structure and function during aging (*Pan et al., 2011*; *Tank et al., 2011*; *Toth et al., 2012*). Structural abnormalities of ALM or PLM appear from the 4th day of adulthood and occur in almost every individual after the 15th day of adulthood (*Toth et al., 2012*). According to our observations, the degree of morphological deformation of the L4 stage *wts-1* was similar to that of the aged wild-type worms on the 15th day of adulthood. These results suggest that loss of *wts-1* leads to structural and functional impairment of TRNs precociously.

### *yap-1* and *egl-44* suppress premature deformation of *wts-1* TRNs

To determine whether *yap-1* acts downstream of *wts-1* in this phenomenon, we assessed genetic interaction of *yap-1* and *wts-1* mutant. In contrast to the *wts-1* mosaic mutant, the *wts-1; yap-1* mutant could survive without intestinal WTS-1 expression as previously reported (*Lee et al., 2019*) and displayed intact neuronal processes without any swelling or branching (*Figure 2A and B*). Introduction of a mutation in *egl-44*, which is the worm homolog of TEAD and functions downstream of *wts-1* along with *yap-1* in the developing intestine (*Lee et al., 2019*), was also sufficient to suppress the structural abnormalities of *wts-1* TRNs (*Figure 2A and B*). To better understand the basic features of structural disintegrity in the *wts-1* mutant and mitigating effects of the *yap-1* mutation, we examined TRNs from each genetic background using electron microscopy. Initially, we observed TRNs of the *wts-1* mutant properly innervate the epidermal layer (*Figure 2—figure supplement 1A*) comparable to the wild-type (*Emtage et al., 2004*). However, TRNs in the *wts-1* mutant showed differences in both the number and morphology of microtubules compared to the wild-type. In wild-type TRNs, multiple microtubules with characteristic spherical shapes were observed (*Figure 2C*). The number of microtubules present in cross sections of control ALM was comparable to the previous report (*Chalfie and Thomson, 1979*) (n = 39, 48, 55, respectively; *Figure 2—figure supplement 1B*). In contrast, in the *wts-1* mutant ALM, the number of microtubules significantly decreased (n = 16, 4, 18, respectively; *Figure 2—figure supplement 1C*) and the morphology was also irregular and non-spherical (*Figure 2C*). Consistent with our finding using a fluorescence marker, the abnormalities in the number and shape of microtubules of TRNs were restored in the *wts-1; yap-1* mutant (n = 24, 30, 44, respectively; *Figure 2—figure supplement 1D*; *Figure 2C*). The loss of *yap-1* also restored the impaired touch responses of the *wts-1* mutant. In comparison with the *wts-1* mutant, touch responses of the *wts-1; yap-1* double mutant were significantly rescued (*Figure 2D and E*).

Furthermore, to rule out the possibility that the ameliorating effect of *yap-1* mutation on neuronal aging was an indirect result of the delayed organismal aging, we measured the lifespan of each mutant. The *wts-1* mutant had a shorter lifespan than the wild-type, suggesting that prematurely deformed neurons of the *wts-1* mutant were probably due to accelerated organismal aging (*Figure 2F*). However, given that the *wts-1; yap-1* mutant with restored neurons had a slightly shorter lifespan than the *wts-1* mutant (*Figure 2F*) and only TRNs were affected in the mutant, premature deformation of the *wts-1* neurons appeared to be a touch neuron-specific event, rather than being associated with whole body.

### The Hpo pathway acts in a cell-autonomous manner to maintain TRNs' integrity

To ascertain whether this function of WTS-1 is cell-autonomous or not, we knocked down WTS-1 selectively in TRNs by injecting tissue-specific RNAi constructs into wild-type animals as previously described (*Esposito et al., 2007*). Sense and antisense constructs corresponding to a region containing kinase domain (sas1) or many exons (sas2) of the *wts-1* gene were expressed under *mec-4* promoter, which is specifically active in all six TRNs (*Figure 3A*). For each construct, three transgenic lines were obtained and TRNs structure of these lines was compared to that of the controls only harboring the transgenic marker, P*unc-122*::RFP. While the control did not have any abnormalities of TRNs, every transgenic lines showed significant disintegration of TRNs such as ectopic neuronal swelling or shortened neuronal process as seen in *wts-1* mutant at L4 stage (*Figure 3B–D*). These neuronal abnormalities induced by TRNs-specific *wts-1* RNAi was consistently observed in 1DA and the penetrance of phenotype was slightly increased (*Figure 3—figure supplement 1A–C*). These results documented that WTS-1 functions in TRNs to maintain intact neuronal structures.

Regarding the mechanism of action of the Hpo pathway in cells, YAP-1 likely acts in the same cell with WTS-1. To clarify this, we rescued YAP-1 activity specifically in TRNs of *wts-1; yap-1* mutants. Touch neuronal expression of YAP-1 using a *mec-4* promoter was sufficient to re-emerge ectopic neuronal swelling in the *wts-1; yap-1* mutants. Approximately 40% of transgenic worms exhibited ectopic swelling and branching on ALM or PLM. In contrast, siblings containing no transgene preserved intact neuronal processes (*Figure 3E and F*).

Overexpression of YAP is commonly observed in many human cancers and is associated with poor prognosis of the diseases (*Liu et al., 2013*; *Yuan et al., 2016*; *Guo et al., 2019*). Inducible overexpression of YAP is sufficient to increase liver size in mice through transcriptional regulation of downstream genes that activate cell proliferation (*Dong et al., 2007*). To see if TRNs-specific overexpression of

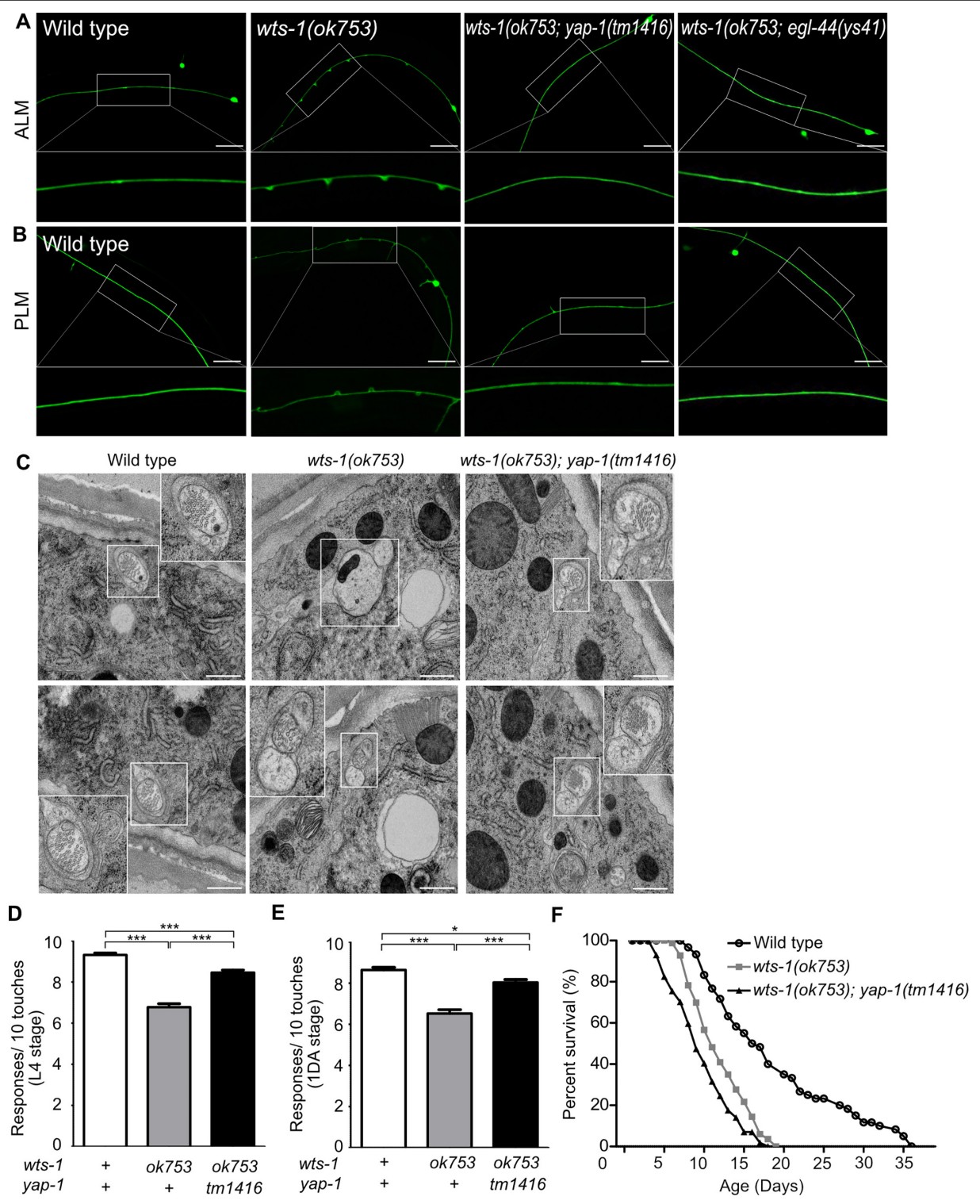

**Figure 2.** *yap-1* or *egl-44* suppresses premature neuronal decline in *wts-1* mutants. (**A, B**) Representative images of (**A**) ALM and (**B**) PLM in wild-type, *wts-1(ok753)*, *wts-1(ok753); yap-1(tm1416)*, and *wts-1(ok753); egl-44(ys41)* at the L4 stage. Loss of *yap-1* or *egl-44* completely restores structural integrity of the *wts-1* mutants. Scale bar = 20 µm. (**C**) Electron microscope images of wild-type, *wts-1(ok753)* and *wts-1(ok753); yap-1(tm1416)* at 1DA. Touch receptor neurons (TRNs) are indicated using boxes. Scale bar = 500 nm. (**D, E**) Touch responses of wild-type, *wts-1(ok753)* and *wts-1(ok753); yap-1(tm1416)* at (**D**) L4 stage and (**E**) 1DA stage. For each strain and each stage, 90 animals were tested. Statistical significance was determined using a one-way ANOVA, followed by Tukey's multiple comparison test. (**F**) Survival curve of wild-type, *wts-1(ok753)* and *wts-1(ok753); yap-1(tm1416)*. Worms were maintained at 20°C, and all lines used for the analysis have *muIs35*.

*Figure 2 continued on next page*

*Figure 2 continued*

The online version of this article includes the following source data and figure supplement(s) for figure 2:

**Source data 1.** Raw data for panels D–F.

**Figure supplement 1.** *yap-1* suppresses structural deformities of *wts-1* mutant neurons.

YAP-1 is sufficient to induce neuronal abnormalities, we overexpressed P*mec-4*::YAP-1 in a wild-type and found that overexpressing YAP-1 causes multiple neuronal defects in a dose-dependent manner (*Figure 3G–J*). Lower expression of YAP-1 (1 ng/μl, final concentration in the injection mixture) rarely affects neuronal structure of ALM or PLM (*Figure 3I and J*). Transgenic worms injected P*mec-4*::YAP-1 at 10 ng/μl final concentration slightly, but not significantly, elevated neuronal defects of TRNs. However, higher expression of YAP-1 in TRNs (50 or 100 ng/μl) resulted in neuronal defects similar to TRNs-specific *wts-1* knockdown (*Figure 3G–J*). It seems that endogenous WTS-1 could inhibit overexpressed YAP-1 at lower concentration along with endogenous YAP-1; however, it fails to inhibit overexpressed YAP-1 at higher levels. These results demonstrated that *wts-1* and *yap-1* act in a cell-autonomous manner to induce structural disintegration of TRNs; thus, proper regulation of YAP-1 activity by WTS-1 is important to maintain neuronal structural integrity.

## Reduced movement alleviates the structural abnormalities of *wts-1*

Neuronal processes contain bundles of microtubules, which are essential structural foundations that protect neurons from mechanical strain-induced damage (*Tang-Schomer et al., 2010*; *Krieg et al., 2017*). We hypothesized that the absence of *wts-1* would affect the microtubule stability in some way; thus, physical stresses generated by movement could induce spontaneous degeneration of neurons. To test this hypothesis, we slowed the movement of *wts-1* mutant via the knockdown of muscle machinery genes using RNAi and observed neuronal morphology. As previously reported (*MacLeod et al., 1977*; *Neumann and Hilliard, 2014*), the knockdown of muscle machinery genes, including a myosin heavy chain gene (*unc-54*), severely impaired organismal movement. Compared with the *wts-1* mutants who were fed with an empty vector (L4440), worms fed with *unc-54* RNAi exhibited uncoordinated movement and some of them were even immobilized (*Figure 4A*). Impaired movement resulting from *unc-54* knockdown also appeared as reduced swimming of animals (*Figure 4E*). Consistent with our hypothesis, *unc-54* RNAi resulted in a significant reduction in neuronal swelling/branching events in ALM as well as PLM (*Figure 4B–D*). The control ALM showed 7.67 lesions on average, whereas ALM treated with *unc-54* RNAi had 4.851 lesions at the L4 stage. For PLM, the events of swelling/branching occurred 6.59 times on average in the controls, whereas 2.96 in the *unc-54* RNAi-fed neurons at the same stage (*Figure 4C*). Additionally, *unc-54* RNAi significantly increased the proportion of intact PLM without any swelling or branching (*Figure 4D*). The knockdown of *unc-15* or *unc-95,* which encodes paramyosin (a structural component of thick filaments) and muscle LIM domain protein, respectively (*Kagawa et al., 1989*; *Broday et al., 2004*), showed movement defects as well and exhibited ameliorative effects on neuronal structures (*Figure 4E and F*). In contrast, *unc-89* and *deb-1* RNAi failed to induce distinguishable movement defects in our experiments (*Figure 4E*). As expected, *unc-89* RNAi did not mitigate neuronal swelling in the *wts-1* mutants and *deb-1* RNAi only had limited effects on ALM, and no effect on PLM (*Figure 4F*). These observations indicate that the severity of movement defects is inversely correlated with the severity of neuronal deformation, and it strongly supports the hypothesis that *wts-1* neurons are vulnerable to physical stresses generated by organismal movements.

## Treatment with colchicine, a microtubule-destabilizing agent, reduced ectopic lesions in *wts-1* neurons

For further understanding of the status of microtubules in the *wts-1* mutant neurons, we evaluated the effects of drugs regulating microtubule stability in the mutant neurons. We transferred the *wts-1* mutants to the drug-containing plates at the L4 stage and scored the number of lesions including ectopic swelling and branching along the processes of their L4 stage-offspring. Interestingly, colchicine, a microtubule-destabilizing agent (*Vandecandelaere et al., 1997*), greatly reduced the morphological abnormalities of the *wts-1* mutant neurons (*Figure 4G*). The drug-untreated ALM showed 5.88 swelling/branching events on a process, whereas the colchicine-treated ALM only had 2.95

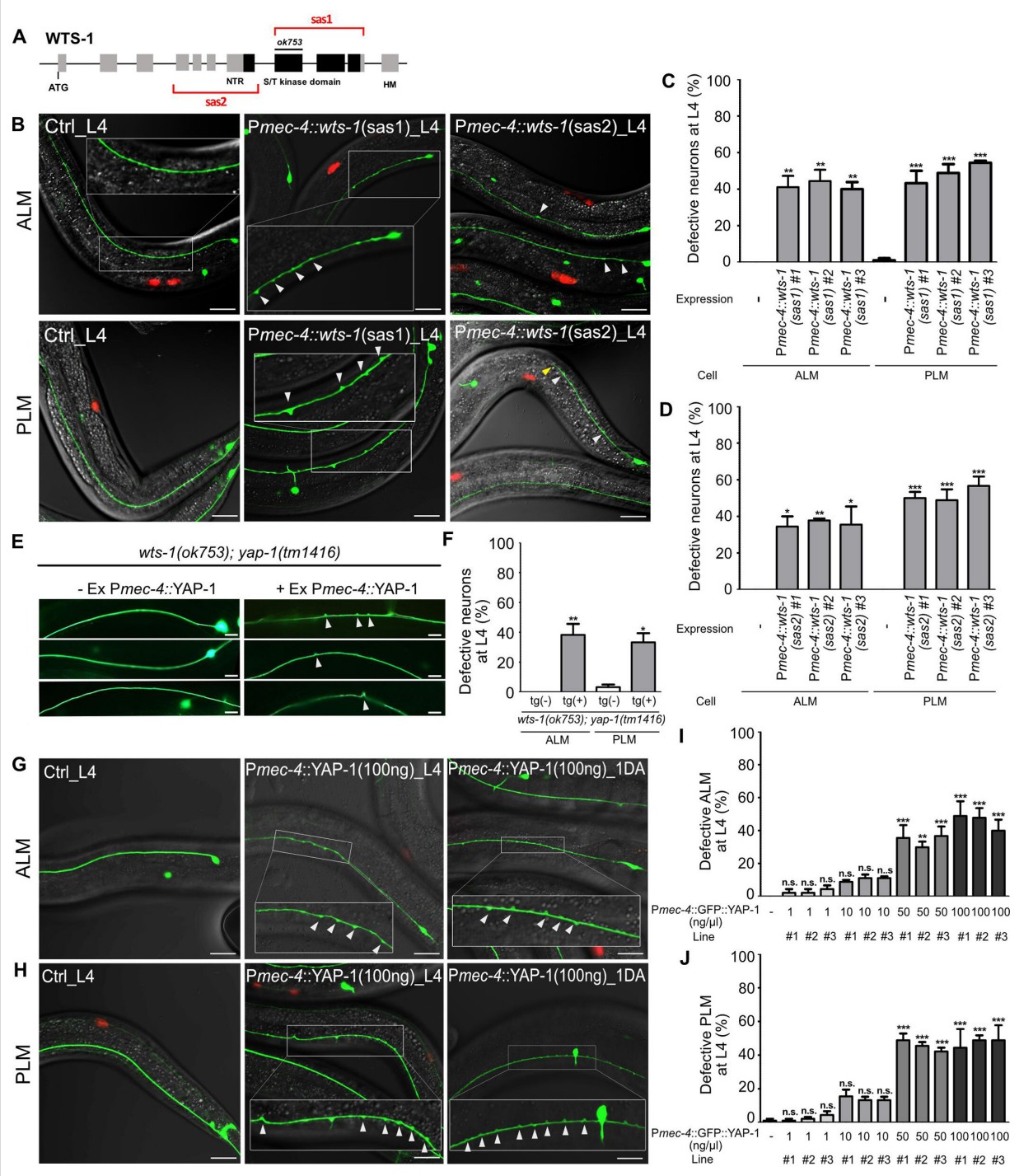

**Figure 3.** *wts-1-yap-1* act in a cell-autonomous manner to maintain touch neuronal integrity. (**A**) The targeting regions for touch neuronal-specific RNAi of *wts-1*. *S*ense and *a*ntisense (sas) genomic fragments were cloned under the touch receptor neurons (TRNs)-specific promoter, P*mec-4*. (**B**) Representative images and (**C, D**) quantified neuronal abnormalities of TRNs in the controls and *wts-1*-knockdowned animals. Both TRNs-specific-sas1- and sas2-based knockdown of *wts-1* efficiently induce neuronal abnormalities as seen in *wts-1* mutant. (**E, F**) Touch neuronal rescue of YAP-1 is sufficient to re-induce neuronal disintegrity in *wts-1(ok753); yap-1(tm1416)* mutant. (**E**) Representative images of TRNs of *wts-1(ok753); yap-1(tm1416)* mutant with the rescue construct or its sibling without transgenes. (**F**) Quantified results. 60 ALM and PLM of transgenic worms and their siblings were observed. (**G, H**) Touch neuronal overexpression of YAP-1 is sufficient to induce neuronal abnormalities in wild-type animals. Final concentration of P*mec-4*::GFP::YAP-1 in the injection mixture is 100 ng/µl. (**I, J**) Quantified neuronal defects of (**I**) ALM and (**J**) PLM induced by overexpression of P*mec-4*::GFP::YAP-1 at each concentration in the injection mixtures. (**C, D, I, J**) 90 ALM and PLM were observed in each of the three independent lines. Neuronal morphology was examined at L4 stage. Statistical significance was determined by a one-way ANOVA followed by Dunnett's multiple

*Figure 3 continued on next page*

*Figure 3 continued*

comparison test. An unpaired *t*-test was used for (**F**). Red: expression of injection marker, P*unc-122*::RFP; white arrowhead: neuronal swelling; yellow arrowhead: shortened process. Scale bar = 20 µm.

The online version of this article includes the following source data and figure supplement(s) for figure 3:

**Source data 1.** Raw data for panels C, D, F, I and J.

**Figure supplement 1.** Cell-autonomous function of *wts-1*.

**Figure supplement 1—source data 1.** Raw data for panels B and C.

swelling/branching. In the case of PLM, colchicine treatment reduced neuronal lesions from 4.41 to 0.89 (*Figure 4H*). Structurally intact neurons were significantly increased by colchicine treatment (*Figure 4I*), and the beneficial effect of colchicine on neuronal structures was consistently observed in TRNs of 1DA offspring (*Figure 4J and K*). Furthermore, treatment with colchicine significantly improved touch responses of the mutants at L4 stage (*Figure 4L*).

In contrast, treatment with paclitaxel, a microtubule-stabilizing agent (*Schiff et al., 1979*; *Vande-candelaere et al., 1997*; *Yvon et al., 1999*), failed to mitigate neuronal abnormalities of the *wts-1* mutants (*Figure 4—figure supplement 1A–C*). Because of the poor cuticle permeability of pacli-taxel, we introduced a mutation in the *bus-17* gene that encodes galactosyltransferase to damage the cuticle of the *wts-1* mutant, as performed in previous studies (*Bounoutas et al., 2009*; *Neumann and Hilliard, 2014*). The number of neuronal lesions was comparable between the paclitaxel-treated *wts-1* neurons and untreated controls (*Figure 4—figure supplement 1B*). These results show that treatment with colchicine, not paclitaxel, lessened the morphological and functional alteration of *wts-1* neurons, suggesting that hyper-stabilized microtubules are responsible for the structural and subsequent func-tional decline of the mutant neurons.

## Hyper-stabilized microtubules are responsible for age-associated neurodegeneration

Next, we questioned whether colchicine could protect neurons from age-associated morphological alteration in wild-type worms. As colchicine treatment had a detrimental effect on the neurodevelop-ment of wild-type unlike in case of the *wts-1* mutant (*Figure 5—figure supplement 1A*), fully devel-oped worms were transferred to colchicine-containing plates at the first day of adulthood (1DA), and neuronal morphologies were monitored in the same generation to assess age-associated morpholog-ical abnormalities. Among morphological alterations of TRNs, bead-like structures on the PLM did not accrue in the aged worms. Beaded PLM was most frequently observed in the young adults (2DA) and gradually decreased as the worms grew older (*Figure 5—figure supplement 1I*). This observation was consistent with the previous study (*Toth et al., 2012*); thus, we excluded beaded PLM from defective neurons in further analyses.

Notably, both ALM and PLM of colchicine-treated worms maintained intact structures even after 20 days of adulthood (20DA) (*Figure 5A–D*). While untreated 20DA had approximately 73.1% and 57.5% defective ALM and PLM, respectively, colchicine-treated worms showed only 23.6% and 7.2% defective ALM and PLM, respectively (*Figure 5C and D*). Colchicine treatment significantly reduced multiple structural abnormalities of TRNs, including somatic outgrowth of ALM and ectopic swelling or branching of PLM (*Figure 5—figure supplement 1B–E*). However, it failed to mitigate other morphological abnormalities such as beaded process of ALM and somatic outgrowth of PLM (*Figure 5—figure supplement 1F–J*). More importantly, colchicine treatment also improved touch responses of aged animals. Whereas there was no difference between touch responses of colchicine-treated and untreated 2DA animals, drug-treated 10DA animals showed significantly higher touch responses compared to the untreated controls (*Figure 5E*). However, touch responses of drug-treated 10DA were still lower than those of drug-treated 2DA (*Figure 5E*). It seems plausible, given that TRNs of colchicine-treated 10DA were more structurally defective than that of colchicine-treated 2DA (*Figure 5C and D*). Considering that the same method of colchicine treatment did not affect worm lifespan (*Figure 5F*), the protective effects of colchicine on the structural and functional integrity of TRNs are unlikely to be derived from the prolonged lifespan of worms.

Next, we examined that hyper-stabilization of neuronal microtubules by paclitaxel treatment could mimic aged phenotype of TRNs. Age-synchronized *bus-17(br2)* mutants, in which drug-permeability

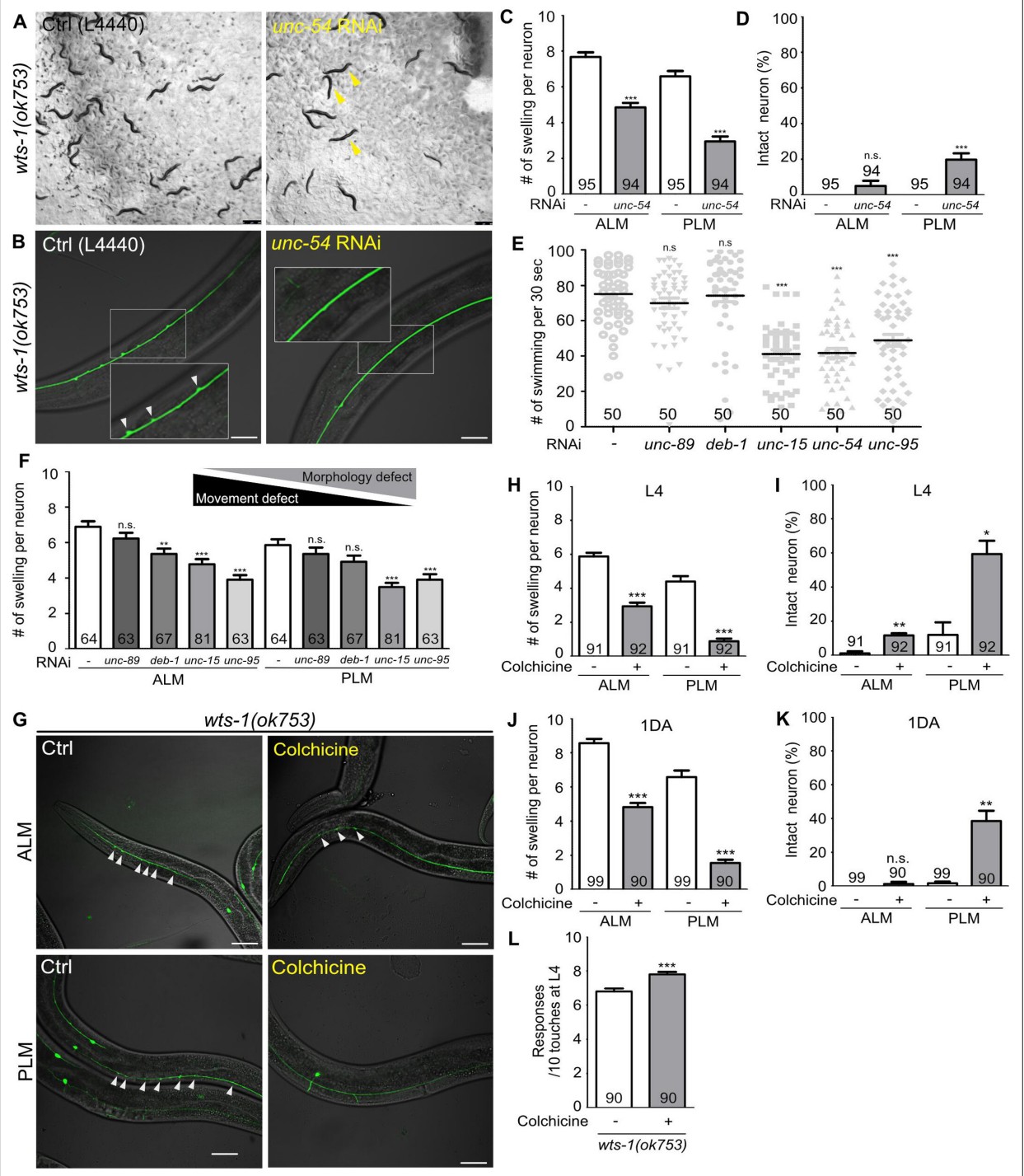

**Figure 4.** Abnormal microtubules are responsible for neuronal disintegrity observed in the *wts-1* mutants. (**A–D**) Reduced movement resulting from *unc-54* knockdown diminishes ectopic swelling or branching in the *wts-1(ok753)* mutant. (**A**) *unc-54* knockdown leads to uncoordinated movement of worms (yellow arrowhead). (**B, C**) *unc-54* knockdown reduces the number of neuronal lesions on the ALM and PLM. (**D**) *unc-54* knockdown slightly, but significantly, increases the percentages of intact PLM and not ALM. Neurons without any structural defects, including swelling or branching, were considered intact. (**E**) Quantified motor deficits in animals knocking down various muscle machinery genes. RNAi against *unc-15, unc-54,* or *unc-95* impairs swimming behavior, whereas *unc-89* and *deb-1* make no differences compared to control. (**F**) Quantified morphological abnormalities of touch receptor neurons (TRNs) in muscle machinery-knockdown animals. Knockdown of *unc-89* or *deb-1* does not alleviate structural decline of both ALM and PLM. (**G–K**) Treatment with colchicine reduces ectopic neuronal swelling and increases the percentages of intact neurons in the *wts-1(ok753)* mutants. F1 progenies grown on the drug-contained plates were scored at (**H, I**) L4 stage or (**J, K**) 1DA. (**L**) Colchicine treatment improves touch responses of the

*Figure 4 continued on next page*

*Figure 4 continued*

*wts-1* mutant. The behavior test was done at L4 stage. Statistical significance was determined using an unpaired *t*-test (**C, D, H, L**) or a one-way ANOVA, followed by Dunnett's multiple comparison test (**E, F**). The total number of cells or animals analyzed is indicated in each column. Asterisks indicate differences from L4440-fed or drug-untreated-control neurons. Ectopic neuronal swelling and branching are labeled with white arrowheads. Scale bar = 20 μm.

The online version of this article includes the following source data and figure supplement(s) for figure 4:

**Source data 1.** Raw data for panels C–F and H–L.

**Figure supplement 1.** Paclitaxel treatment fails to mitigate *wts-1* neuronal defects.

**Figure supplement 1—source data 1.** Raw data for panels B and C.

was enhanced as mentioned, were transferred to paclitaxel-containing plates at L4 stage and their neuronal morphologies were scored at 5DA stage of the same generation. Although Treatment with paclitaxel did not affect PLM structure, it significantly increased neuronal defects of ALM (*Figure 5— figure supplement 1K and L*). Paclitaxel induced beaded processes, somatic outgrowth, and cell body abnormalities of ALM, as well as bipolar growth, which is reported to be induced by paclitaxel. Taken together, hyper-stabilized microtubules could be responsible for age-related neuronal degeneration, and neuronal status of the *wts-1* mutant was similar to that of aged neurons.

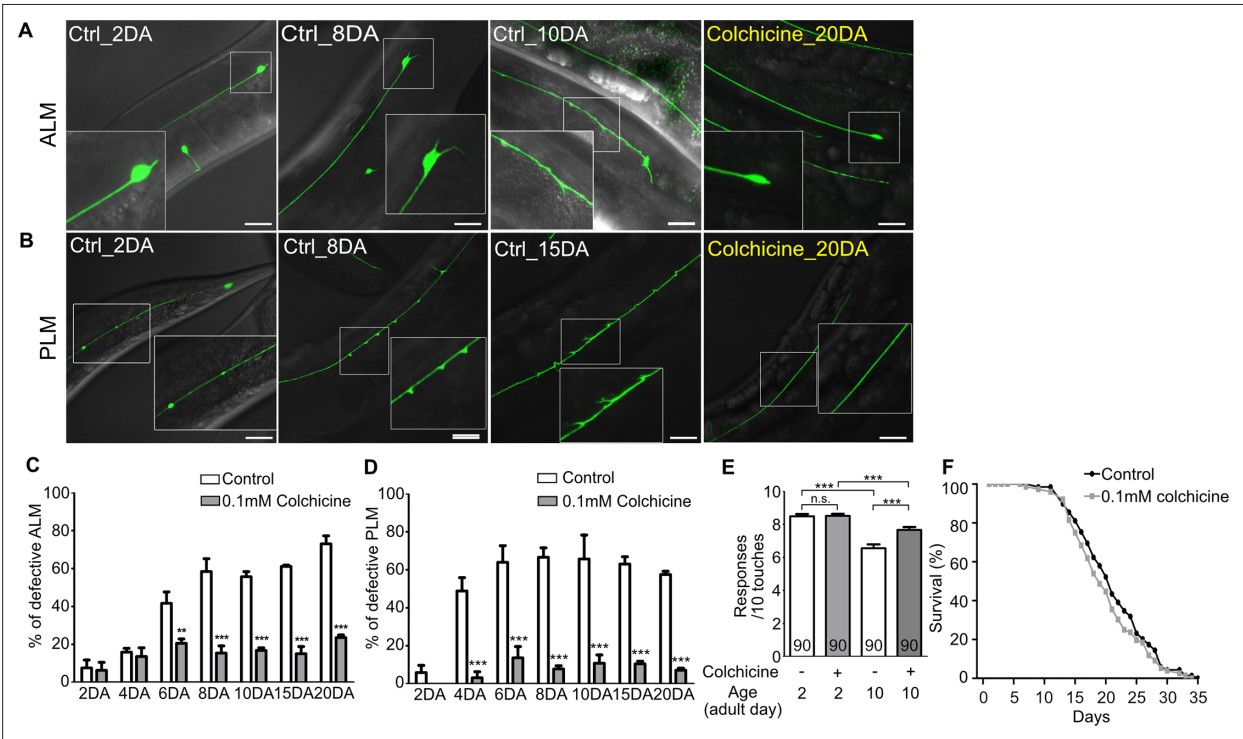

**Figure 5.** Hyper-stabilized microtubules might be responsible for age-associated morphological deformation of touch receptor neurons. (**A, B**) Representative images of (**A**) ALM and (**B**) PLM of colchicine-treated or untreated wild-type worms. The age of worms is indicated in the image. (**C, D**) Quantified defects of (**C**) ALM or (**D**) PLM in colchicine-treated or untreated worms. Age-synchronized worms were transferred to drug-containing plates on the 1DA and phenotypes were scored on every 2nd, 4th, 6th, 8th, 10th, 15th, and 20th day of adulthood. At every time point and in each group, 20 neurons were scored per one experiment and the experiments were repeated three times. (**E**) Colchicine treatment alleviates impaired touch responses of aged animals. Touch responses were scored at 2DA and 10DA. Statistical significance was determined by a two-way ANOVA, followed by Bonferroni's post-test (**C, D**) or a one-way ANOVA with Bonferroni's multiple comparison test (**E**). (**F**) Survival curves of colchicine-treated and untreated control animals. Scale bar = 20 μm.

The online version of this article includes the following source data and figure supplement(s) for figure 5:

**Source data 1.** Raw data for panels C–F.

**Figure supplement 1.** Colchicine treatment affects touch receptor neurons (TRNs) morphology.

**Figure supplement 1—source data 1.** Raw data for panels B–J and L.

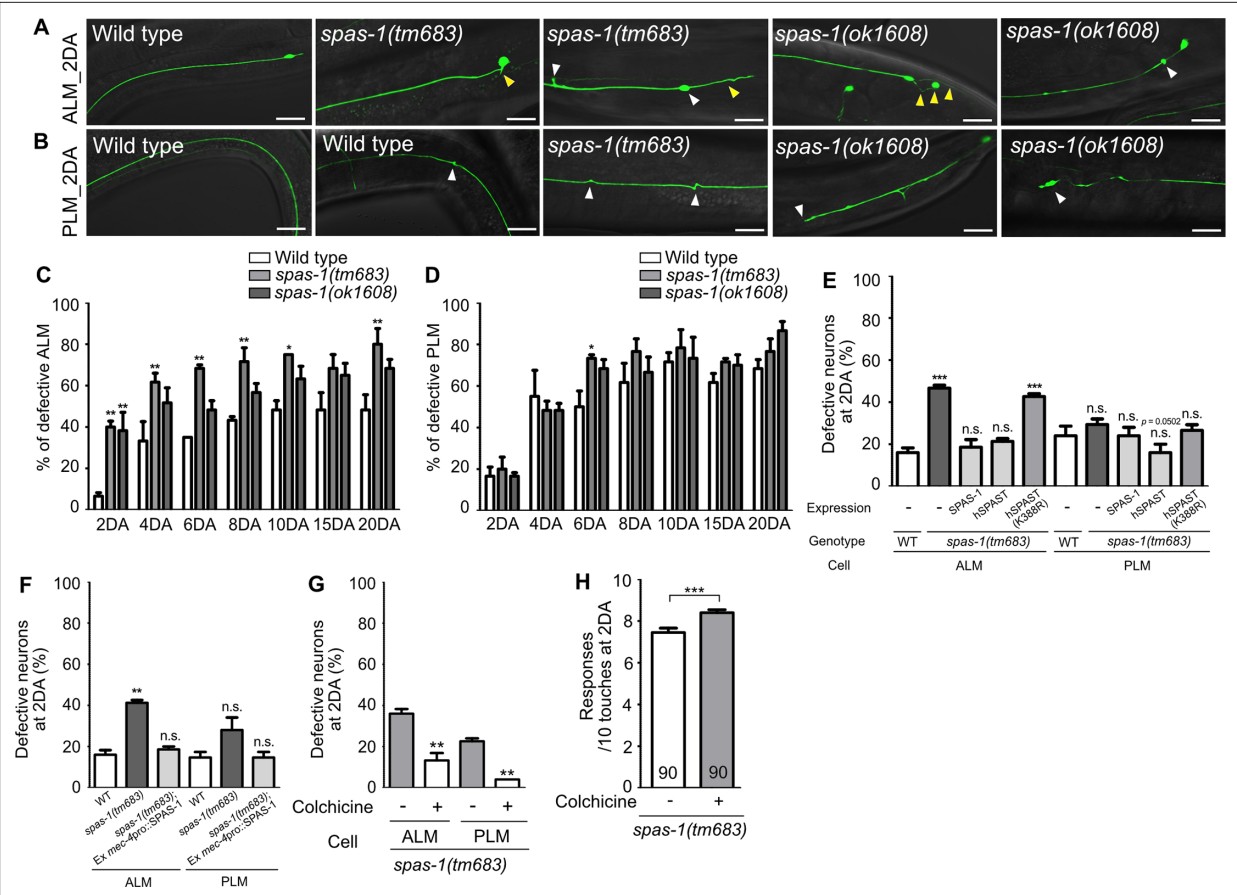

**Figure 6.** Loss of *spas-1*, a microtubule-severing enzyme, results in premature structural decline. (**A, B**) Representative images of defective touch receptor neurons (TRNs) of *spas-1* mutant on the 2DA. Both deletion mutants, *tm683* and *ok1608,* display age-associated morphological alterations of ALM or PLM including ectopic swelling and branching on the neuronal process (white arrowhead), somatic outgrowth and irregular shape of the cell body (yellow arrowhead) precociously. (**C, D**) Quantified results of structural defects of ALM or PLM in *spas-1* mutants. At every time point, 20 neurons were scored and the experiments were repeated three times. (**E**) Results of the rescue experiment of premature neuronal degeneration of *spas-1(tm683)* with SPAS-1, human SPAST(wt), and human SPAST(K388R). In all cases, the *C. elegans spas-1* promoter was used to induce the C-terminally mCherry tagged transgene. (**F**) Touch neuronal-specific expression of SPAS-1 was sufficient to rescue neuronal defects of ALM. (**E, F**) Neuronal morphology were scored at 2DA. For each rescue construct, three independent transgenic lines were observed that yielded similar results and the results of one line are presented. (**G, H**) Quantified (**G**) structural defects of TRNs or (**H**) touch responses of colchicine-treated and untreated *spas-1(tm683)* mutant. Analyses were done at 2DA (N = 30/1 experiment, repeated three times). Statistical significance was determined using a two-way ANOVA, followed by Bonferroni's multiple comparison test (**C, D**), a one-way ANOVA with Dunnett's multiple comparison test (**E, F**) or with Turkey's multiple comparison test (**G**). Unpaired *t*-test was used in (**H**). Scale bar = f20 μm.

The online version of this article includes the following source data and figure supplement(s) for figure 6:

**Source data 1.** Raw data for panels C–H.

**Figure supplement 1.** *spas-1* also acts cell-autonomously to maintain neuronal morphology of touch receptor neurons (TRNs).

## Loss of *spas-1*, a putative microtubule-severing enzyme, accelerates structural decline of TRNs

Spastin, an ATP-dependent microtubule-severing enzyme, is known to cause a human neurodegenerative disease, hereditary spastic paraplegia (HSP) (*Hazan et al., 1999*; *Svenson et al., 2001*). Hyperstabilized microtubules by the loss of spastin activity have been considered as a main contributor of disease pathology (*Sherwood et al., 2004*; *Evans et al., 2005*). SPAS-1, the worm homolog of spastin, possesses a microtubule-severing activity and is expressed in the TRNs (*Matsushita Ishiodori et al., 2007*; *Brown and El Bejjani, 2017*). We investigated whether hyper-stabilization of microtubules via genetic perturbation of *spas-1* could cause premature structural defects of TRNs as observed in *wts-1* mutant. We found that the *spas-1* mutants exhibit premature deformation of TRNs. From the

second day of adulthood (2DA), the mutant ALM displayed an accelerated onset of neuronal deformity (*Figure 6A and C*). Although the frequency of defective PLM was not significantly increased in the mutants except in *spas-1(tm683)* at 6DA (*Figure 6D*), both the deletion mutants *tm683* and *ok1608* exhibited more various and severe structural deformation of PLM than controls on the 2DA and 6DA (*Figure 6B*, *Figure 6—figure supplement 1A*). Ours and others' observation showing touch neuronal expression of SPAS-1 suggest cell-autonomous function of SPAS-1 (*Figure 6—figure supplement 1B*; *Brown and El Bejjani, 2017*). Consistent with this hypothesis, expression of SPAS-1 under *mec-4* promoter, as well as *spas-1* own promoter, restored premature deformation of the *spas-1* mutant ALM (*Figure 6E and F*). Human SPAST expression under the control of the *C. elegans spas-1* promoter displayed similar rescue effects, whereas the loss-of-function mutant (K388R), which is associated with disease pathology (*Fonknechten et al., 2000*), failed to restore the structural integrity of the mutant ALM (*Figure 6E*). In the case of PLM, human spastin expression or P*mec-4*-driven Ce_SPAS-1 expression slightly, but not significantly, reduced the structural deformation of the mutants (p=0.0502 and 0.1161, respectively), and other rescue constructs did not affect the PLM structures (*Figure 6E and F*).

Next, we investigated that pharmacological destabilization of neuronal microtubules by colchicine treatment could ameliorate structural defects of *spas-1* mutant. As done in *wts-1* mutant background, F1 progenies of animals grown on the drug-contained plate were observed. We found that colchicine treatment not only rescued structural defects of *spas-1* mutant, but also increased touch responses of the mutant at 2DA (*Figure 6G and H*).

In human spastin, K388R mutation completely abrogates the ATP binding in the ATPase region and microtubule-severing activity (*Evans et al., 2005*). Taken together, microtubule-severing activity of SPAST is likely conserved in *C. elegans* SPAS-1, and it is important to protect neuronal structures from gradual deformation. Moreover, the fact that the loss of microtubule-severing enzyme led to accelerated neuronal deformation also supports that hyper-stabilized microtubules are responsible for the age-associated structural decline of neurons.

## Loss of microtubule-stabilizing genes *dlk-1* and *ptl-1* delayed premature deformation of *wts-1* neurons

We next studied the genetic interaction between *wts-1* and several genes that are known to regulate microtubule stability, particularly those expected to act in TRNs. We selected six potential microtubule-stabilizing genes (*atat-2*, *dlk-1*, *mec-17*, *mig-2*, *ptl-1,* and *ptrn-1*) and five microtubule-destabilizing genes (*kin-18*, *rho-1*, *unc-33*, *spas-1*, and *elp-1*). A *wts-1; eri-1* mutant was generated by genetic mating to enhance the efficiency of neuronal RNAi (*Kennedy et al., 2004*), and we evaluated lesions in the TRNs of worms fed with the RNAi vector of target genes. RNAi against microtubule-stabilizing genes, except for *atat-2* and *mec-17*, could reduce the structural deformation of *wts-1; eri-1* mutant neurons (*Figure 7—figure supplement 1B*). *atat-2* RNAi slightly alleviated ectopic lesions in ALM and PLM. *mec-17* RNAi failed to reduce PLM lesions but increased ALM lesions. In contrast, none of the microtubule-destabilizing genes reduced ectopic lesions of the mutant neurons. Among the candidate genes, *rho-1* RNAi resulted in early larval arrest, making it infeasible to test; thus, RNAi experiments were performed for only four genes, which failed to reduce the structural deformation of *wts-1* mutant neurons (*Figure 7—figure supplement 1A*).

To address the genetic interaction of microtubule-stabilizing genes and *wts-1* in detail, we constructed double mutants with *wts-1* and *dlk-1*, *ptl-1*, and *ptrn-1*. Both *wts-1; dlk-1* and *wts-1; ptl-1* displayed improved TRNs structures. The total number of ectopic lesions was significantly reduced, and the proportion of intact neurons without any deformed structures was also highly increased (*Figure 7A–C*). In the case of *ptrn-1*, neuroprotective effects of the gene knockdown were not reproduced in the double mutant (*Figure 7—figure supplement 1C and D*). The fact that the loss of microtubule-stabilizing genes mitigates the structural deformation of the *wts-1* also supports the finding that the *wts-1* mutants have highly stabilized microtubules which are responsible for the structural deformation.

*ptl-1* encodes the sole *C. elegans* homolog of the microtubule-associated protein tau (MAPT) (*McDermott et al., 1996*). In mammals, tau is predominantly localized to the neuronal axons and promotes microtubule assembly and stability; moreover, mutations in the MAPT locus are highly associated with several neurodegenerative diseases, such as FTD-17 (*Goedert and Spillantini, 2000*). In *C. elegans*, the loss of *ptl-1* itself results in premature structural disintegration of TRNs (*Chew et al.,*

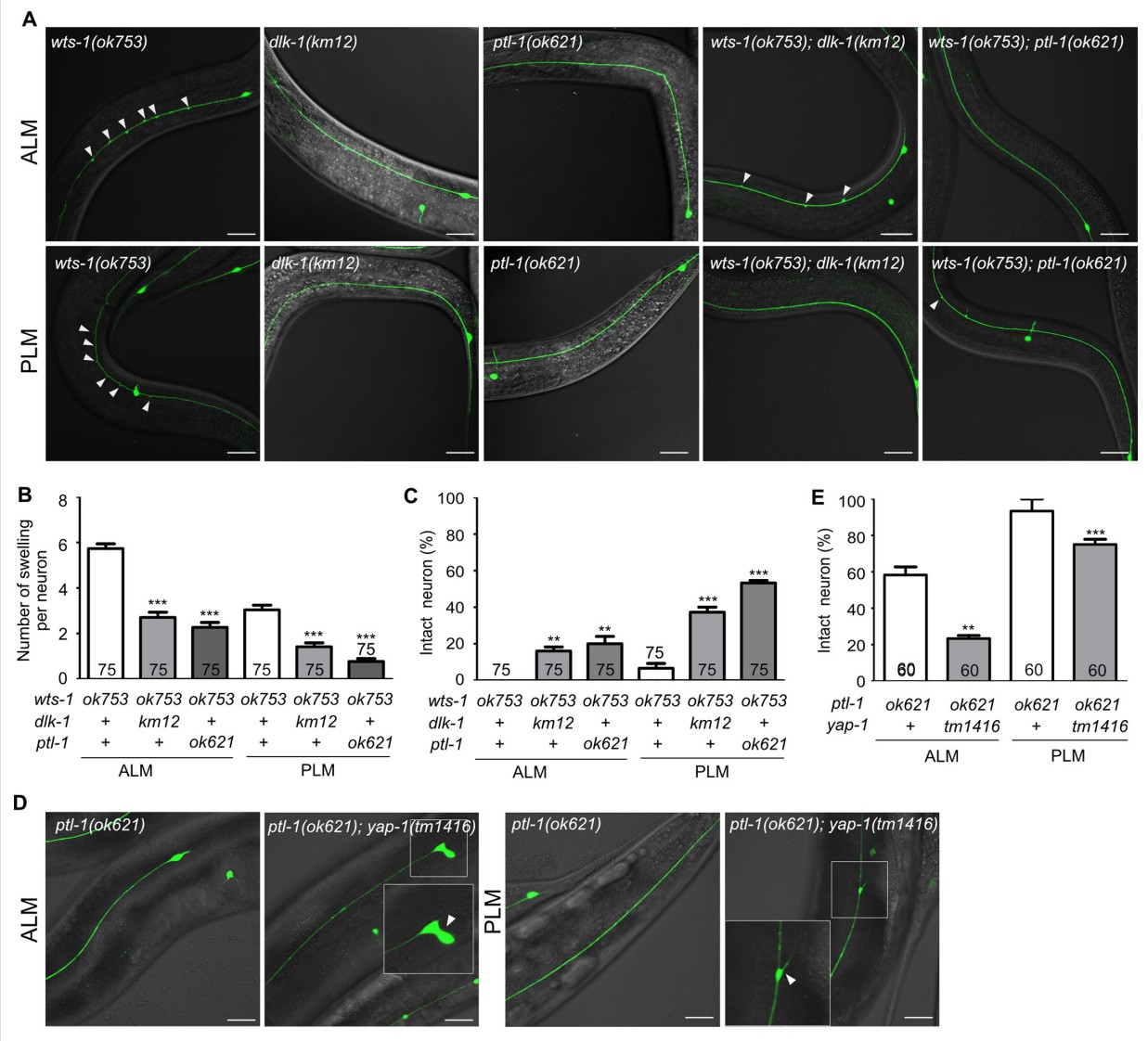

**Figure 7.** *wts-1-yap-1* affect neuronal integrity possibly by modulating microtubule stability. (**A–C**) Loss of *dlk-1* or *ptl-1* significantly mitigates the structural deformation of *wts-1*-mutant neurons. (**A**) Representative images of ALM (upper panels) and PLM (lower panels) of *wts-1(ok753)*, *dlk-1(km12)*, *ptl-1(ok621)*, *wts-1(ok753); dlk-1(km12)*, and *wts-1(ok753); ptl-1(ok621)* at L4 stage. Ectopic lesions are labeled with arrowhead. (**B**) Average number of ectopic lesions per neuronal process of each strain. (**C**) Percentage of intact neurons of each mutant. Loss of *dlk-1* or *ptl-1* protects touch receptor neurons (TRNs) of the *wts-1* from premature deformation. (**D, E**) Loss of *yap-1* worsens the neuronal deformation as seen in the *ptl-1* mutant. (**D**) Unlike *ptl-1(ok621)* single mutant, *ptl-1(ok621); yap-1(tm1416)* double mutant exhibits severe deformation in ALM and PLM, such as the irregular shape of cell body and ectopic branching of the neuronal process. (**E**) Percentage of undamaged touch neurons in *ptl-1(ok621)* and *ptl-1(ok621); yap-1(tm1416)*. (**A–E**) Neurons were analyzed at the L4 stage. The number of scored neurons is indicated in each column. Statistical significance was determined using a one-way ANOVA, followed by Dunnett's multiple comparison test. Asterisks indicate differences from the *wts-1* or the *ptl-1* single mutant. Scale bar = 20 μm.

The online version of this article includes the following source data and figure supplement(s) for figure 7:

**Source data 1.** Raw data for panels B, C, and E.

**Figure supplement 1.** Microtubule-destabilizing genes, not stabilizing genes, mitigate morphological deformations of *wts-1* mutant neurons.

**Figure supplement 1—source data 1.** Raw data for panels A, B and D.

*2013*). Given these observations with the *wts-1*; *ptl-1* mutants and the revealed functions of MAPT, the structural decline seen in the *ptl-1* mutants is probably due to the microtubule destabilization, unlike those observed in the *wts-1* mutants. Consistent with this assumption, the loss of *yap-1* did not lessen structural deformation of *ptl-1*, even increased the structural abnormalities of *ptl-1* (***Figure 7D and E***). On the fifth day of adulthood, *ptl-1* mutant exhibited 41% defective ALM and 6% of defective

PLM. In the *ptl-1; yap-1* mutant, cells with irregular shape, ectopic branching, or somatic outgrowth were highly increased; thus, 76% ALM and 25% PLM showed structural abnormalities (*Figure 7D and E*). Loss of *yap-1* could probably lead to microtubule destabilization and has a redundant effect on microtubule stability with microtubule destabilization due to *ptl-1* loss.

## Discussion

In this study, we showed that the loss of *wts-1*, the core kinase of the Hpo pathway, results in post-natal deformation of TRNs. Although *wts-1* mutants had intact neurons at the beginning, they gradually exhibited structurally and functionally declined TRNs. The detailed observation of morphological alteration in *wts-1* mutant suggests that the features of defective TRNs in the *wts-1* mutant closely resemble those of aged TRNs (*Toth et al., 2012*). We also defined that both *yap-1* and *egl-44* act as downstream of *wts-1* and that TRNs-specific rescue of YAP-1 was sufficient to re-induce the neuronal deformation of *wts-1; yap-1* double mutant. Neuronal defects induced by knockdown of *wts-1* or overexpression of YAP-1 selectively in TRNs of wild-type animals also proved a cell-autonomous function of the Hpo pathway in TRNs. Given that *daf-2*/IGF signaling modulates whole organismal aging, premature decline of the *wts-1* mutants is more local and restricted in TRNs. These observations say that the Hpo pathway of *C. elegans* has a neuroprotective effect in differentiated neurons in a cell-autonomous manner, restricting YAP-1 from triggering premature neuronal decline.

In addition to its extensively studied roles in early development and tumorigenesis, dysregulation of the Hpo pathway has been implicated in aging and pathologies of the nervous system (*Sahu and Mondal, 2020*; *Gogia et al., 2021*). Hyper-activation of the pathway components such as MST1 and the consequent inhibition of YAP have been noted in several neurodegenerative disease models (*Matsumoto et al., 2014*; *Yamanishi et al., 2017*; *Mueller et al., 2018*; *Tanaka et al., 2020*). In these models, the inactivation of the pathway or the activation of YAP ameliorates neuronal cell death; thus, the manipulation of the Hpo pathway has been regarded to prevent pathologies associated with neuronal cell death. Our results showing that the proper inhibition of YAP-1 by an upstream WTS-1 is required to maintain neuronal integrity appear to contradict these previous observations. However, alterations that occur in the aged brain or the *wts-1* mutants are structural alterations or failures in structural maintenance, which are different from neuronal cell death that occurs in neurodegenerative diseases. This could explain the conflicting function of YAP-1 in terms of neuroprotection. Moreover, some examples present the detrimental effects of uncontrolled YAP. Dysregulation of the pathway and ectopic activation of YAP have been observed in patients with Alexander disease, which is a rare neurodegenerative disease that results in progressive neuronal degeneration based on the loss of myelin (*Wang et al., 2018*). Further, the activation of YAP also occurred in Müller cells during retinal degeneration (*Hamon et al., 2017*). The present study shows that proper regulation of YAP is essential to maintain neuronal integrity and adds important information to the research on anti-aging roles of the Hpo pathway after development.

Genetic and pharmacological approaches suggest that defective, hyper-stabilized microtubules in the *wts-1* mutant were responsible for premature deformation of neurons, possibly in a 'wear-to-tear fashion' because of the mechanical strains generated by organismal movements. Reduced movement or treatment with colchicine had significant ameliorating effects on neuronal deformation of the *wts-1* mutant. Colchicine binds irreversibly to tubulin dimers and prevents the addition of tubulin dimers to the fast-growing ends of microtubule (*Ravelli et al., 2004*). It has been widely used to cure acute gouty arthritis and familial Mediterranean fever (FMF) (*Dinarello et al., 1974*; *Zemer et al., 1974*; *Zemer et al., 1986*). Our study demonstrated that colchicine reduced the severity of neuronal lesions and even improved neuronal function in the *wts-1* mutant TRNs. Moreover, colchicine treatment on fully developed animals largely reduced age-related failures in neuronal structures and functions during normal aging. This supports the idea that neuronal deformation seen in the *wts-1* mutants is similar to that in aged organism, and, more importantly, hyper-stabilization of neuronal microtubules would be a promising target to cure age-associated neuronal deformation.

These observations are unexpected because microtubule-destabilizing agents mimic several neuronal degenerative disease phenotypes, and microtubule-stabilizing agents such as paclitaxel and epothilone D exhibit neuroprotective effects in the pathological contexts (*Zhang et al., 2005*; *Shemesh and Spira, 2011*; *Zhang et al., 2012*). Some reports have shown that colchicine has detrimental effects on cognitive function in animal disease models specifically leading to cholinergic

neuronal loss (*Goldschmidt and Steward, 1982*; *Veerendra Kumar and Gupta, 2002*). In contrast, a study in older patients with FMF showed that long-term colchicine treatment could protect against cognitive decline in patients (*Leibovitz et al., 2006*) and the neuroprotective functions of colchicine also have been reported (*Pratt et al., 1994*; *Salama et al., 2012*). In the case of HSP and paclitaxel-induced peripheral neuropathy, hyper-stabilization of microtubule could be a cause of neurodegenerative diseases (*Cavaletti et al., 1995*; *Hazan et al., 1999*; *Trotta et al., 2004*; *Evans et al., 2005*; *Lee and Swain, 2006*; *Scripture et al., 2006*; *Gornstein and Schwarz, 2014*). Mutated spastin is the most common cause of HSP, a hereditary, neurodegenerative disease that affects the upper motor neurons (*Hazan et al., 1999*). Similar to the neuroprotective effects of colchicine in the *wts-1* or aged neurons, vinblastine, a microtubule-destabilizing agent, rescues axonal swellings in the HSP models (*Fassier et al., 2013*). Consistently, the loss of *spas-1*, the worm homolog of spastin, led to premature deformation of TRNs, and treatment with colchicine alleviated premature decline of *spas-1* mutant TRNs in both structure and function. Additionally, the fact that impaired spastin activity in several animal models results in sparser microtubule array at axons (*Sherwood et al., 2004*; *Wood et al., 2006*) provides a possible link between hyper-stabilization and the sparseness of microtubules we observed. These shared properties among *wts-1* mutant neurons, aged neurons, and HSP models strongly support that the hyper-stabilized microtubules are responsible for the structural and functional failures of neurons in both senescent and pathological conditions. It will be needed to determine that colchicine has more general neuroprotective effects on age- or diseases-associated decline in other animal models and other microtubule-destabilizing agents show similar beneficial effects.

Mutations in several tubulin proteins, microtubule-binding proteins such as tau, and failures in nerve attachment to the epidermis have been reported to induce similar deformities in TRNs, as seen in the *wts-1* mutant. In several mutants of *mec-1*, which encodes an ECM protein, TRNs fail to separate from body-wall muscles and they are prematurely degenerated (*Pan et al., 2011*). In contrast to the *mec-1* mutant neurons, TRNs of the *wts-1* mutant appeared to normally dissociate from the muscles and innervate the epidermal layer. Uncontrolled *yap-1* may affect the vesicular transport of membrane proteins by altering plasma membrane polarity in the same way it acts in the intestine. Since we were unable to identify transcriptional targets of YAP-1 associated with these phenomena, the detailed mechanism of how the activated YAP-1 triggers microtubule hyper-stabilization in TRNs cell-autonomously remains to be explored. The unique 15-protofilament microtubules in TRNs (*Chalfie and Thomson, 1982*), which may contribute to their vulnerability to age-related deformation, could be influenced by transcriptional regulation through YAP-1. Otherwise, YAP-1 may have broader effects in neuronal microtubules and other characteristics of TRNs, such as their anatomical localization near the cuticle or their long projections along the body axis, which could also affect age-related deformation. Given that YAP is known to regulate transcription of secretory proteins, such as TGF-β ligand (*Lee et al., 2016*), an extracellular factor possibly influences touch neuronal microtubules in a non-cell-autonomous manner. Despite these limitations in understanding the underlying mechanism, the phenotype of the *wts-1* mutant appears much earlier, and its phenotypic penetrance is notably higher compared with any other known mutant. Moreover, to the best of our knowledge, this is one of the first reports to show the impact of the signaling pathway on the specific neuronal aging. As we have shown in this study, defective TRNs of the *wts-1* mutant could provide a neuronal model to investigate the potential therapeutic targets for improving neuronal aging that occurs either prematurely or normally.

## Materials and methods
### Worm maintenance and strains

Worms were maintained at 20°C as previously described (*Brenner, 1974*), unless noted otherwise. To visualized TRNs, we isolated *muIs35 [Pmec-7::GFP + lin-15(+)] V* from CF1192 *egl-27(n170) II; muIs35 [Pmec-7::GFP + lin-15(+)] V* by outcrossing four times with N2 wild-type. This strain was used as a wild-type control and *muIs35* was transferred to each mutant background to track touch neurons in all experiments, except the *spas-1* experiments. Since *spas-1* gene is localized at the same chromosome with *muIs35*, CF702 *muIs32[Pmec-7::GFP + lin-15(+)] II* was used as control and transferred in each mutant background. Exact deletion site of *wts-1(ok753)* was confirmed by PCR. It starts +4025

from ATG start codon and ends +4610. It also has a 'C' insertion after deletion (581 bp deletion and 1 insertion). Detailed descriptions of the strains used in this study are available in *Supplementary file 1*.

## Molecular biology

To construct P*opt-2::wts-1*::GFP, a 2.3 kb promoter of *opt-2* and the genomic DNA of *wts-1* were inserted into the pPD95.75 vector. To generate P*mec-4*::GFP::*yap-1*, a 0.5 kb promoter of *mec-4* and the genomic DNA of *yap-1* were inserted into the pCF150 with the gateway cloning method. To visualize dopaminergic, GABAergic, and cholinergic neurons of worms, 2 kb promoters of *dat-1*, *unc-47*, and *cho-1* were respectively fused to GFP using a standard fusion PCR method.

To construct P*mec-4::wts-1*(sas1), a 0.5 kb promoter of *mec-4* and either a sense or antisense fragment corresponding to exons 8–10 of *wts-1* were individually inserted into the pPD117.01 vector. For P*mec-4::wts-1*(sas2), a 0.5 kb promoter of *mec-4* and either a sense or antisense fragment exons 4–7 of *wts-1* were inserted into the pPD117.01.

To construct P*spas-1::spas-1(c)*::mCherry, a 0.5 kb promoter of *spas-1* and the *spas-1* cDNA were inserted into the pJL1050 vector. To generate human spastin rescue constructs, a 0.5 kb promoter of *spas-1* and cDNA of either *SPAST(wt)* or *SPAST(K388R)* were inserted into the pJL1050. cDNA were PCR-amplified from mApple-M1 Spastin (Addgene # 134461) and mApple-M1 SpastinK388R (Addgene #134463), respectively. For P*mec-4::spas-1(c)*::mCherry, a 0.5 kb promoter of *mec-4* and *spas-1* cDNA were inserted into the pJL1050. Primer sequence information is available upon request.

## Fluorescence microscopy

To monitor neuronal morphology, a fluorescence microscopy (Axioplan2, Carl Zeiss, Inc) was used. All fluorescence images were acquired using the confocal microscope (ZEISS LSM700, Carl Zeiss, Inc) and ZEN software (Carl Zeiss, Inc).

## RNAi

For the *unc-54* RNAi construct, 2000 bps from sixth exon region was cloned into the L4440 vector. For *unc-95*, *deb-1*, *mig-2*, *mec-17*, and *rho-1*, we used constructs from Marc Vidal libraries. The other genes were from the J. Ahringer libraries. RNAi feeding experiments were done by standard methods. 4–6 L4 worms were transferred to plates with bacteria expressing RNAi constructs, and neuronal morphology was scored at L4 stage of F1 generations.

## Fluorescence microscopy and phenotype scoring

To monitor neuronal morphology, a fluorescence microscope (Axioplan2, Carl Zeiss, Inc) was used. All fluorescence images were acquired using the confocal microscope (ZEISS LSM700, Carl Zeiss, Inc) and ZEN software (Carl Zeiss, Inc) To score mutant phenotype, 30 neuronal cells of stage-synchronized *wts-1(ok753)* mutants were observed for each stage, and the experiments were repeated three times. To gain age-synchronized animals, 50 *wts-1* mutants were transferred to the new plate at the first day of adulthood and were removed leaving laid eggs 2 hours later. Since *wts-1(ok753)* develops slower than wild-type control, L1 stage worms 16 hours after egg-laying, L2 worms 40 hours later, L3 worms 54 hours later and L4 worms 72 hours later, and 1DA worms after 96 hours were observed. Morphological abnormalities not found in the L4 stage-wild-type worms were considered as defects. To quantify phenotype severity of the mutant, the total number of ectopic swelling and branching on a neuronal process was measured.

## Gentle touch test

Touch sensitivity was tested by stroking the animal with an eyebrow hair attached to a toothpick. To test anterior touch response, we stroked around the pharynx of the worm moving forward. In response of touch, the worm moving backward was scored as 'touch sensitive'. In case of posterior touch response, the worm moving backward was tested and tail of the worm was touched.

## Drug treatment

Colchicine (Sigma-Aldrich, C9754) was added as dry powder into hot agar at 0.1 mM concentration. For *wts-1* mutant, 4–5 L4 worms were transferred to plates containing or not containing colchicine. 4 days later, neuronal morphology was scored in their L4 progenies. To monitor colchicine effects on

normal aging, we transferred about 300 synchronized 1-day adult animals to each plate containing or not containing the drug and transferred to new plates every 1–2 days to remove F1 progenies. At 2, 4, 6, 8, 10, 15, and 20 days in their adulthoods, 20 touch neuronal cells were observed. In case of paclitaxel (Sigma-Aldrich, T7402), DMSO was used as the solvent because paclitaxel is poorly soluble in water. Final concentration of paclitaxel is 1 µM. Plates only containing DMSO were used as the control. The experiment was performed in the same way as the colchicine treatment into *wts-1*.

## Lifespan measurement

Lifespan was measured as the standard method with some modifications. For each strain or condition, 100 L4 worms were transferred to the plates (25 worms per one plate, four plates) and living worms were transferred into new plates for every 1~3 days to remove F1 progenies. Worms that did not to respond to touch using a platinum wire were scored to be dead. Animals that ruptured from vulva bursting, bagged, crawled off, or burrowed into the plates were excluded from the analysis. Every measurement was repeated three times and yielded similar results. One representative experimental result was shown. Statistical analyses were performed using OASIS2 (https://sbi.postech.ac.kr/oasis2/; *Han et al., 2016*), and significance was determined by log-rank (Mantel–Cox) test.

## Electron microscopy

*C. elegans* animals were overlaid with 20% bovine serum albumin in M9 buffer and immediately high-pressure frozen using a Leica EM HPM100 apparatus (Leica, Austria). Animals were transferred to the freeze substitution apparatus (Leica EM AFS) under liquid nitrogen into a solution containing 2% osmium tetroxide and 2% water in acetone. Samples were maintained at –90°C for 100 hours, slowly warmed to –20°C (5°C per hour) and maintained for 20 hours, and slowly warmed to 0°C (6°C per hour). Three washes with cold acetone were carried out at 0°C, and samples were embedded in Embed-812 (EMS, USA). After polymerization of the resin at 60°C for 36 hours, serial sections were cut with a diamond knife on an ULTRACUT UC7 ultramicrotome (Leica, Austria) and mounted on formvar-coated grids. Sections were stained with 4% uranyl acetate for 10 minutes and lead citrate for 7 minutes. They were observed using a Tecnai G2 Spirit Twin transmission electron microscope (FEI Company, USA).

## Acknowledgements

We thank the Caenorhabditis Genetics Center and the National BioResource Project for providing strains, Julie Ahringer for RNAi plasmids. We also thank Andrew Fire for the worm-expressing vectors. The plasmids containing human SPAST genes were gifts from Jennifer Lippincott-Schwartz (Addgene plasmid #134461, #134463). This work was supported by the Samsung Science and Technology Foundation under project Number SSTF-BA1501-52 and the National Research Foundation of Korea grant funded by the Korean government (MEST) (2019R1A6A1A10073437). H Lee was supported by a scholarship for basic researches, Seoul National University, Seoul, Korea.

## Additional information

### Funding

| Funder | Grant reference number | Author |
|---|---|---|
| National Research Foundation of Korea | 2019R1A6A1A10073437 | Junho Lee |
| Samsung Science and Technology Foundation | SSTF-BA1501-52 | Hanee Lee<br>Christine H Chung<br>Junho Lee |

The funders had no role in study design, data collection and interpretation, or the decision to submit the work for publication.

## Author contributions
Hanee Lee, Conceptualization, Data curation, Formal analysis, Validation, Investigation, Visualization, Methodology, Writing – original draft, Writing – review and editing; Junsu Kang, Conceptualization, Investigation, Writing – review and editing; Sang-Hee Lee, Investigation, Visualization, Methodology; Dowoon Lee, Christine H Chung, Investigation; Junho Lee, Conceptualization, Supervision, Funding acquisition, Writing – review and editing

## Author ORCIDs
Hanee Lee ⓘ https://orcid.org/0000-0003-1511-5997
Junsu Kang ⓘ https://orcid.org/0000-0001-5286-5426
Junho Lee ⓘ https://orcid.org/0000-0002-6421-1195

Joint Public Review: https://doi.org/10.7554/eLife.102001.3.sa1
Author response https://doi.org/10.7554/eLife.102001.3.sa2

# Additional files

## Supplementary files
Supplementary file 1. Summary of *C. elegans* strains used in this study.

MDAR checklist

## Data availability
All data generated or analysed during this study are included in the manuscript and supporting files.

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
