## [Editor Report · eLife Assessment]

In their **valuable** study, Lee et al. explore a role for the Hippo signaling pathway, specifically wts-1/LATS and the downstream regulator yap, in age-dependent neurodegeneration and microtubule dynamics using *C. elegans* mechanosensory neurons as a model. The authors demonstrate that disruption of wts-1/LATS leads to age-associated morphological and functional neuronal abnormalities, linked to enhanced microtubule stabilization, and show a genetic connection between yap and microtubule stability. Overall, the study employs robust genetic and molecular approaches to reveal a **convincing** link between the Hippo pathway, microtubule dynamics, and neurodegeneration.

---

## [Referee Report · Joint Public Review]

The Lee et al. study has been revised in response to reviewer comments. It presents a valuable investigation into the role of the Hippo signaling pathway (specifically wts-1/LATS and yap) in age-dependent neurodegeneration and microtubule dynamics in *C. elegans* TRNs. The authors convincingly demonstrated that disruption of wts-1/LATS leads to age-associated neuronal abnormalities and enhanced microtubule stabilization, with a genetic link to yap. While the study was praised for its well-conducted and well-controlled approaches, reviewers raised concerns about the specificity of the Hippo pathway's effects to TRNs, the correlation of Hpo signaling decline in TRNs with age, and the mechanistic link between Hpo-mediated gene expression and microtubule regulation. The authors addressed the TRN specificity by suggesting the unique microtubule structure of these neurons might contribute to their susceptibility. They acknowledged the difficulty in detecting Hpo signaling decline specifically in aged TRNs but noted increased YAP-1 nuclear localization in other tissues. Importantly, the authors provided evidence suggesting that YAP-TEAD-mediated transcriptional regulation is responsible for neuronal degeneration, as loss of yap-1 or egl-44 restored the wts-1 mutant phenotype. However, the specific transcriptional targets of YAP-1 regulating microtubule stability remain unidentified, representing a key limitation. The authors also discussed the possibility of non-cell-autonomous effects of YAP-1 and offered explanations for the seemingly moderate impairment of the touch response despite structural damage. Finally, they attributed the shorter lifespan of wts-1 and wts-1; yap-1 mutants to roles of wts-1 beyond TRNs and potential synergistic effects of yap-1. Overall, the study provides significant insights into the Hippo pathway's role in neuronal aging and microtubule dynamics, while acknowledging remaining mechanistic gaps.

---

## [Author Response]

The following is the authors’ response to the original reviews.

**Public Reviews:**

**Reviewer #1 (Public review):**
Summary:In this manuscript, the authors investigate the role of microtubule dynamics and its effects on neuronal aging. Using *C. elegans* as a model, the authors investigate the role of evolutionarily conserved Hippo pathway in microtubule dynamics of touch receptor neurons (TRNs) in an age-dependent manner. Using genetic, molecular, behavioral, and pharmacological approaches, the authors show that age-dependent loss of microtubule dynamics might underlie structural and functional aging of TRNs. Further, the authors show that the Hippo pathway specifically functions in these neurons to regulate microtubule dynamics. Specifically, authors show that hyperactivation of YAP-1, a downstream component of the Hippo pathway that is usually inhibited by the kinase activity of the upstream components of the pathway, results in microtubule stabilization and that might underlie the structural and functional decline of TRNs with age. However, how the Hippo pathway regulates microtubule dynamics and neuronal aging was not investigated by the authors.Strengths:This is a well-conducted and well-controlled study, and the authors have used multiple approaches to address different questions.Weaknesses:There are no major weaknesses identified, except that the effect of the Hippo pathway seems to be specific to only a subset of neurons. I would like the authors to address the specificity of the effect of the Hippo pathway in TRNs, in their resubmission.

Although our genetic experiments, including TRNs-specific rescue/overexpression of YAP-1 and knockdown of WTS-1, strongly suggest that a cell-autonomous function of WTS-1-YAP-1 axis in TRNs, the Hpo pathway could have broader roles in neuroprotection. While this pathway may regulate microtubules stability in multiple neurons, other characteristics of TRNs, such as their anatomical localization near the cuticle or their long projections along body axis, could contribute to their susceptibilities to age-related deformation. Otherwise, the Hpo pathway may be truly TRNs-specific. TRNs have unique microtubules in both terms of composition and structure. Among nine α-, six β-tubulin genes in *C. elegans*, one α-tubulin (*mec-12*) and one β-tubulin (*mec-7*) showed highly enriched expression in TRNs [1, 2] and TRNs contain special 15-protofilament microtubule structure, while all other neurons in *C. elegans* have 11-protofilament microtubules [3]. Transcriptional regulation through YAP-1 may affect the specific microtubule structure of TRNs, leading to premature neuronal deformation. We have included this in the discussion section of the revised manuscript.

**Reviewer #2 (Public review):**
Summary:This study examines a novel role of the Hpo signaling pathway, specifically of wts-1/LATS and the downstream regulator of gene expression, yap, in age-related neurodegeneration in *C. elegans* touch-responsive mechanosensory neurons, ALM and PLM. The study shows that knockdown or deletion of wts-1/LATS causes age-associated morphological abnormalities of these neurons, accompanied by functional loss of touch responsiveness. This is further associated with enhanced, abnormal, microtubule stabilization in these neurons.Strengths:This study examines a novel role of the Hpo signaling pathway, specifically of wts-1/LATS and the downstream regulator of gene expression, yap, in age-related neurodegeneration in *C. elegans* touch-responsive mechanosensory neurons, ALM and PLM. The study shows that knockdown or deletion of wts-1/LATS causes age-associated morphological abnormalities of these neurons, accompanied by functional loss of touch responsiveness. This is further associated with enhanced, abnormal, microtubule stabilization in these neurons. Strong pharmacological and especially genetic manipulations of MT-stabilizing or severing proteins show a strong genetic link between yap and regulation of MTs stability. The study is strong and uses robust approaches, especially strong genetics. The demonstrations on the aging-related roles of the Hpo signaling pathway, and the link to MTs, are novel and compelling. Nevertheless, the study also has mechanistic weaknesses (see below).Weaknesses:Specific comments:(1) The study demonstrates age-specific roles of the Hpo pathway, specifically of wts-1/LATS and yap, specifically in TRN mechanosensory neurons, without observing developmental defects in these neurons, or effects in other neurons. This is a strong demonstration. Nevertheless, the study does not address whether there is a correlation of Hpo signaling pathway activity decline specifically in these neurons, and not other neurons, and at the observed L4 stage and onwards (including the first day of adulthood, 1DA stage). Such demonstrations of spatio-temporal regulation of the Hpo signaling pathway and its activation seem important for linking the Hpo pathway with the observed age-related neurodegeneration. Can this age-related response be correlated to indeed a decline in Hpo signaling during adulthood? Especially at L4 and onwards? It will be informative to measure this by examining the decline in wts1 as well as yap levels and yap nuclear localization.

As described above, we have included possible explanations for the specificity of the Hpo pathway in TRNs. Since components of the Hpo pathway are expressed in various tissues, including the intestine and hypodermis, this pathway could have broader neuroprotective roles across multiple neurons. Alternatively, it could function in TRNs. Given that the TRNs possess unique microtubules in both structure and composition, and that Hpo pathway has crucial roles in microtubule stability regulation, the roles of the Hpo pathway may indeed be TRNs-specific. As we described in the manuscript, our observations, along with those of others, indicate that neuronal deformation of TRNs begins around the 4th day of adulthood. Additionally, the degree of morphological deformation in *wts-1* mutants at the L4 stage is comparable to that of aged wild-type worms on the 15th day of adulthood. Therefore, to assess the functional decline of WTS-1 *or nuclear localization of YAP-1*, observations should begin in 4-day-old animals. Using fluorescence-tagged YAP-1 under the *mec-4* promoter, we couldn’t detect a significant increase in nuclear YAP-1 in TRNs of 4-day-old adult. Additionally, we were unable to assess YAP-1 intercellular localization in older animals, such as 10-day-old animals, possibly due to the small cell size of neurons or morphological alteration along with aging of TRNs. Although we did not detect functional decline of WTS-1 or increased nuclear YAP-1 in TRNs, nuclear localization of YAP-1 increases with age in other tissues, such as the intestine and hypodermis (Author response image. 1). This may result from inactivation of the Hippo (Hpo) pathway, an indirect consequence of structural and functional decline—such as tissue stiffness associated with aging—or a combination of both. Additionally, given that morphological deformation of TRNs appears to begin around fourth day of adulthood, nuclear localization of YAP-1 in the intestine and hypodermis seems to have a later onset and be more moderate. It is possible that YAP-1 nuclear localization in TRNs occurs earlier or that other factors contribute early-stage touch neuronal deformation.

**Author response image 1. sa2fig1:** Quantification of the proportion of worms exhibiting nuclear localization of YAP-1. We used GFP-tagged YAP-1 driven by its own 4 kb promoter. A total of 90 animals were observed each day.

(2) The Hpo pathway eventually activates gene expression via yap. Although the study uses robust genetic manipulations of yap and wts-1/LATS, it is not clear whether the observed effects are attributed to yap-mediated regulation of gene expression (see 3).

Given that the neuronal deformation in the *wts-1* mutant was completely restored by the loss of *yap-1* or *egl-44*, it strongly suggests that YAP-TEAD-mediated transcriptional regulation is responsible for the premature neuronal degeneration of the *wts-1* mutant. However, in this study, we were unable to identify specific transcriptional target genes associated with these phenomena, which represents a limitation of our research (please see below).

(3) The observations on the abnormal MT stabilization, and the subsequent genetic examinations of MT-stability/severing genes, are a significant strength of the study. Nevertheless, despite the strong genetic links to yap and wts-1/LATS, it is not clear whether MT-regulatory genes are regulated by transcription downstream of the Hpo pathway, thus not enabling a strong causal link between MT regulation and Hpo-mediated gene expression, making this strong part of the study mechanistically circumstantial. Specifically, it will be good to examine whether the genes addressed herein, for example, Spastin, are transcriptionally regulated downstream of the Hpo pathway. This comment is augmented by the finding that in the wts-1/ yap-1 double mutants, MT abnormality, and subsequent neuronal morphology and touch responses are restored, clearly indicating that there is an associated transcriptional regulation

If the target genes of YAP-1 are not identified, it will be difficult to fully understand how YAP-1 regulates microtubule stability. Microtubule-stabilizing genes, whose knockdown alleviates *wts-1* mutant neuronal deformation, could be potential transcriptional targets of YAP-1. Among these genes, PTRN-1 and DLK-1 contain MCAT sequences (CATTCCA/T), a well-conserved DNA motif recognized by the TEAD transcription factor, in their promoters near the transcription start site (TSS). We hypothesized that the expression of fluorescence-tagged reporters of promoter regions containing these MCAT sequences would be enhanced in the absence of *wts-1* activity. Although both reporters were expressed in TRNs, they did not show significant changes in the *wts-1* mutant background. We also focused on *spv-1*, a worm homolog of ARHGAP29, which negatively regulates RhoA. YAP is known to modulate actin cytoskeleton rigidity through transcriptional regulation of ARHGAP29 [4]. The promoter of *spv-1* contains 2 MCAT sequences and loss of *spv-1* mitigated neuronal deformation of the *wts-1* mutant. However, reporters of promoter regions containing MCAT sequences only weakly expressed in the process of TRNs. More importantly, ectopic expression of dominant-negative form of rho-1/rhoA did not lead to significant deformation of TRNs. While YAP typically functions as a transcriptional co-activator, it has also been reported to repress target gene expression, such as DDIT4 and Trail, in collaborated with TEAD transcriptional factor [5]. As a reviewer pointed out, *spas-1* might be transcriptionally repressed by *yap-1,* given that its loss leads to premature deformation of TRNs. However, since the phenotype of the *spas-1* mutant has a later onset than the *wts-1* mutant and is relatively restricted to ALM, we excluded it from our candidate gene search. Despite extensive genetic approaches, we were unable to establish a strong causal link between YAP-1 and the regulation of microtubule stability. Unbiased screenings, such as tissue-specific transcriptome analysis, may help address the remaining questions. We have outlined the limitations of this study in the discussion section of the revised manuscript.

Other comments:(1) The TRN-specific knockdown of wts-1 and yap-1 is a clear strength. Nevertheless, these do not necessarily show cell-autonomous effects, as the yap transcription factor may regulate the expression of external cues, secreted or otherwise, thus generating non-cell autonomous effects. For example, it is known that yap regulates TGF-beat expression and signaling.

In the absence of LATS1/2 activity, activated YAP has been reported to drive biliary epithelial cell lineage specification by directly regulating TGF-β transcription during and after liver development [6]. Even when functioning in an autocrine manner, TGF-β can exhibit non-cell autonomous effects. While it primarily acts on the same cell that secretes it, some molecules may also affect neighboring cells, leading to paracrine effects. Additionally, TGF-β can modify the extracellular matrix (ECM), indirectly affecting surrounding cells. Similarly, if YAP regulates transcription of secretory protein in TRNs, the resulting extracellular factors or surrounding cells may influence touch neuronal microtubules in a non-cell-autonomous manner. Although our genetic data strongly suggest a cell-autonomous function of WTS-1-YAP-1 in TRNs, we could not exclude the possibility that YAP-1 functions non-cell-autonomously, as we were unable to identify its transcriptional targets. We have included this in the discussion section of the revised manuscript.

(2) Continuing from comment (3) above, it seems that many of the MT-regulators chosen here for genetic examinations were chosen based on demonstrated roles in neurodegeneration in other studies. It would be good to show whether these MT-associated genes are directly regulated by transcription by the Hpo pathway.

As we described above, several MT-associated genes­­, such as *ptrn-1*, *dlk-1* and *spv-1*, contain MCAT sequences in their promoter and their knockdown alleviated *wts-1*-induced neuronal deformation. These genes were tested to determine whether they were directly regulated by WTS-1-YAP-1. Based on our findings, we concluded that they were unlikely to be regulated by the Hpo pathway in TRNs.

(3) The impairment of the touch response may not be robust: it is only a 30-40% reduction at L4, and even less reduction at 1DA. It would be good to offer possible explanations for this finding.

As pointed out by the reviewer, the impairment of touch responses of *wts-1* mutants showed an approximately 33% reduction at both L4 and 1DA compared to age-matched wild-type animals. At the L4 stage, control worms responded to nearly every gentle touch (94%), whereas *wts-1* mutants responded to only 60% of stimuli. By 1DA, control worms exhibited slightly decline in touch responses compared to L4 (82.5%), whereas *wts-1* mutants displayed more pronounced impairment (55.7%) (Fig 1E). Regarding the severity and frequency of structural degeneration of *wts-1* mutant at both stages, it appears to be relatively moderate. As we noted in the manuscript, our observations, along with those of others, indicate that structural abnormalities in ALM and PLM neurons begin to appear around the fourth day of adulthood and progressively worsen as the worms age [7]. In a previous study, Tank et al. categorized day 10-aged worms into two groups based on their movement ability and then assessed structural deformation in each animal to determine whether structural and functional degeneration of TRNs were correlated. In this same group of animals, they examined the gentle touch response and found that animals responded to gentle touch 46 ± 5.1 %, 84 ± 12.2 %, respectively [8]. It could be said that, on average, day 10 animals had 65% touch response on average, which is consistent with our observation in day 10 animals (Fig. 5E, 56.3%). Given these observations, the function of TRNs of *wts-1* mutant or aged animals appears to be preserved despite severe structure failures. The gentle touch response evokes an escape behavior in which animals quickly move away from the stimulus; thus proper touch responses are essential for avoiding predators and ensuring survival. It has been reported to be necessary for evading fungal predation, such as escaping from a constricting hyphal ring [9]. Given that the gentle touch response is crucial for survival, its function is likely well preserved despite structural abnormalities, such as age-related deformation.

**Reviewer #1 (Recommendations for the authors):**
Major comments:(1) Why is the effect of the Hippo pathway on microtubule dynamics specific to TRNs? Is it the structure of TRNs that makes them prone to the effects of age-dependent decline in microtubule dynamics? The authors are advised to discuss it in their resubmission.

As described above, we have included possible explanations for the tissue specificity of the Hpo pathway in TRNs and the vulnerability of TRNs to age-associated decline in the discussion section of the revised manuscript.

(2) The authors are advised to explain the shorter life span of wts-1; yap-1 double mutants (with restored TRNs) compared to wts-1 single mutants in Figure 2F. The life span of yap-1 single mutants should be included in Figure 2F. Further, based on the data, the shorter lifespan of wts-1 mutants cannot be attributed to abnormal TRNs as the lifespan of wts-1; yap-1 double mutants is even shorter. The authors are advised to explain the shorter life span of wts-1 mutants compared to wild-type controls.

*wts-1* is known to be involved in various developmental processes, including the maintenance of apicobasal polarity in the intestine, growth rate control, and dauer formation [10-12]. Since WTS-1 activity is restored in the intestine of the mutant used for lifespan measurement, the shorter lifespan of the *wts-1* mutant may result from the loss of WTS-1 in tissues other than the intestine. Although we were unable to include lifespan data for the *yap-1* mutant, recent studies indicate that the *yap-1(tm1416)* mutant or *yap-1* RNAi treated worms exhibit a shortened lifespan [13, 14]. Thus, our data showing a slightly shorter lifespan of the *wts-1; yap-1* mutant compared with the *wts-1* mutant may result from the synergistic action of *yap-1* and *yap-1*-independent downstream factors of *wts-1*. While this study does not provide an explanation for the shortened lifespan of *wts-1* or *wts-1; yap-1* mutants, the fact that the *wts-1; yap-1* double mutant with restored TRNs still have a shorter lifespan compared with the *wts-1* mutant strongly suggests that premature deformation of the *wts-1* neurons appear to be a touch neuron-specific event, rather than being associated with whole body, as described in the manuscript..

Minor comments:(1) In the abstract, please provide definitions for LATS and YAP. Authors can mention that LATS is a kinase and YAP a transcriptional co-activator in the Hippo pathway.(2) In the last paragraph on page 9, change "these function" to "this function", and change "knock-downed" to "knocked down".(3) On page 10, paragraph 2, change "regarding the action mechanism" to "regarding the mechanism of action".(4) On page 11, paragraph 1, change "endogenous WTS-1 could inhibits" to "endogenous WTS-1 could inhibit".(5) On page 16, paragraph 1, change "consistent to the hypothesis" to "consistent with this hypothesis".(6) Overall, the paper is well written. However, there is still room to improve the language and diction used by the authors.

We have revised all minor comments suggested by the reviewer in the revised manuscript.

References

(1) Hamelin M, Scott IM, Way JC, Culotti JG. The mec-7 beta-tubulin gene of *Caenorhabditis elegans* is expressed primarily in the touch receptor neurons. EMBO J. 1992;11(8):2885-93. Epub 1992/08/01. doi: 10.1002/j.1460-2075.1992.tb05357.x. PubMed PMID: 1639062; PubMed Central PMCID: PMCPMC556769.

(2) Fukushige T, Siddiqui ZK, Chou M, Culotti JG, Gogonea CB, Siddiqui SS, et al. MEC-12, an alpha-tubulin required for touch sensitivity in *C. elegans*. J Cell Sci. 1999;112 (Pt 3):395-403. Epub 1999/01/14. doi: 10.1242/jcs.112.3.395. PubMed PMID: 9885292.

(3) Chalfie M, Thomson JN. Structural and functional diversity in the neuronal microtubules of *Caenorhabditis elegans*. J Cell Biol. 1982;93(1):15-23. Epub 1982/04/01. doi: 10.1083/jcb.93.1.15. PubMed PMID: 7068753; PubMed Central PMCID: PMCPMC2112106.

(4) Qiao Y, Chen J, Lim YB, Finch-Edmondson ML, Seshachalam VP, Qin L, et al. YAP Regulates Actin Dynamics through ARHGAP29 and Promotes Metastasis. Cell Rep. 2017;19(8):1495-502. Epub 2017/05/26. doi: 10.1016/j.celrep.2017.04.075. PubMed PMID: 28538170.

(5) Kim M, Kim T, Johnson RL, Lim DS. Transcriptional co-repressor function of the hippo pathway transducers YAP and TAZ. Cell Rep. 2015;11(2):270-82. Epub 2015/04/07. doi: 10.1016/j.celrep.2015.03.015. PubMed PMID: 25843714.

(6) Lee DH, Park JO, Kim TS, Kim SK, Kim TH, Kim MC, et al. LATS-YAP/TAZ controls lineage specification by regulating TGFbeta signaling and Hnf4alpha expression during liver development. Nat Commun. 2016;7:11961. Epub 2016/07/01. doi: 10.1038/ncomms11961. PubMed PMID: 27358050; PubMed Central PMCID: PMCPMC4931324.

(7) Toth ML, Melentijevic I, Shah L, Bhatia A, Lu K, Talwar A, et al. Neurite sprouting and synapse deterioration in the aging *Caenorhabditis elegans* nervous system. J Neurosci. 2012;32(26):8778-90. Epub 2012/06/30. doi: 10.1523/JNEUROSCI.1494-11.2012. PubMed PMID: 22745480; PubMed Central PMCID: PMCPMC3427745.

(8) Tank EM, Rodgers KE, Kenyon C. Spontaneous age-related neurite branching in *Caenorhabditis elegans*. J Neurosci. 2011;31(25):9279-88. Epub 2011/06/24. doi: 10.1523/JNEUROSCI.6606-10.2011. PubMed PMID: 21697377; PubMed Central PMCID: PMCPMC3148144.

(9) Maguire SM, Clark CM, Nunnari J, Pirri JK, Alkema MJ. The *C. elegans* touch response facilitates escape from predacious fungi. Curr Biol. 2011;21(15):1326-30. Epub 2011/08/02. doi: 10.1016/j.cub.2011.06.063. PubMed PMID: 21802299; PubMed Central PMCID: PMCPMC3266163.

(10) Cai Q, Wang W, Gao Y, Yang Y, Zhu Z, Fan Q. Ce-wts-1 plays important roles in *Caenorhabditis elegans* development. FEBS Lett. 2009;583(19):3158-64. Epub 2009/09/10. doi: 10.1016/j.febslet.2009.09.002. PubMed PMID: 19737560.

(11) Kang J, Shin D, Yu JR, Lee J. Lats kinase is involved in the intestinal apical membrane integrity in the nematode *Caenorhabditis elegans*. Development. 2009;136(16):2705-15. Epub 20090715. doi: 10.1242/dev.035485. PubMed PMID: 19605499.

(12) Lee H, Kang J, Ahn S, Lee J. The Hippo Pathway Is Essential for Maintenance of Apicobasal Polarity in the Growing Intestine of *Caenorhabditis elegans*. Genetics. 2019;213(2):501-15. Epub 20190729. doi: 10.1534/genetics.119.302477. PubMed PMID: 31358532; PubMed Central PMCID: PMCPMC6781910.

(13) Teuscher AC, Statzer C, Goyala A, Domenig SA, Schoen I, Hess M, et al. Longevity interventions modulate mechanotransduction and extracellular matrix homeostasis in *C. elegans*. Nat Commun. 2024;15(1):276. Epub 2024/01/05. doi: 10.1038/s41467-023-44409-2. PubMed PMID: 38177158; PubMed Central PMCID: PMCPMC10766642.

(14) Saul N, Dhondt I, Kuokkanen M, Perola M, Verschuuren C, Wouters B, et al. Identification of healthspan-promoting genes in *Caenorhabditis elegans* based on a human GWAS study. Biogerontology. 2022;23(4):431-52. Epub 2022/06/25. doi: 10.1007/s10522-022-09969-8. PubMed PMID: 35748965; PubMed Central PMCID: PMCPMC9388463.